# An Analytical Framework for Investigating Trade-Offs between Reservoir Power Generation and Flood Risk

**Lin Zhang [1,2], Jay R. Lund [2], Wei Ding [1,*], Xiaoli Zhang [3], Sifan Jin [1], Guoli Wang [1] and Yong Peng [1]**

[1] School of Hydraulic Engineering, Dalian University of Technology, Dalian 116024, China
[2] Center for Watershed Sciences, Department of Civil and Environmental Engineering, University of California, Davis, CA 95616, USA
[3] School of Water Resources, North China University of Water Resources and Electric Power, Zhengzhou 450045, China
*   Correspondence: weiding@dlut.edu.cn

**Abstract:** Converting floodwater into power without increasing flood risk is critical for energy-stressed regions. Over the past decades, numerous methods have been proposed to solve this problem. However, few studies have investigated the theoretical explanation of the trade-offs between power generation and flood risk. This study establishes an analytical framework to derive optimal hedging rules (OHR) and explains the economic insights into flood risk reduction and power generation improvement. A two-stage model based on the concept of dynamic control of carryover storage (DCCS) was developed as part of the framework, considering forecast uncertainty and risk tolerance. The results illustrated that hedging and trade-offs between power generation and flood risk during DCCS only occurs when the forecasted inflow and forecast uncertainty are within certain ranges, beyond which there is no hedging and trade-offs analysis; either power generation or flood risk becomes the dominant objective. The OHR was divided into three cases under different levels of forecast uncertainty and risk tolerance. Compared to forecast uncertainty, downstream risk tolerance plays a more important role in determining which case of the OHR is adopted in real-world operations. The analysis revealed what and how intense trade-offs are between power generation and flood risk under different scenarios of forecasted inflow, forecast uncertainty, and risk tolerance. The framework serves as a guideline for less abundant water resources or energy-stressed areas of operational policy. Nierji Reservoir (located in northeast China) was taken as a case study to illustrate the analysis, and the application results showed that OHR increases the average annual power generation by 4.09% without extra flood risk compared to current operation rules.

**Keywords:** optimal hedging rules; two-stage model; trade-offs; flood risk; power generation

## 1. Introduction

The energy crisis and climate change catalyze the demand for renewable energy sources [1]. Optimal hydropower reservoir operation is highly valued because it effectively promotes renewable power generation, which has great environmental advantages for reducing the emissions of fossil fuel combustion [2–4]. Reservoir operation is a decision-making process that seeks the optimal storage volume for multiple objectives [5–7]. For the operation during flood season, flood control is the prime concern and requires a reduction in storage to accommodate the incoming floods and minimize downstream damage caused by excess flow, which conflicts with power generation [8,9]. To solve this problem, significant effort has been expended to elevate storage volume to increase power generation [10–13], in which the dynamic control of carryover storage (DCCS) is proven an effective method [14–16]. For instance, Jiang et al. [17] proposed a credibility-based fuzzy chance-constrained model, and fuzzy simulation technology was used to optimize the dynamic control bound. Liu et al. [18] focused on the multi-objective optimal

scheduling of the dynamic control of the flood-limited water level for cascade reservoirs based on a multi-objective evolutionary algorithm to obtain the trade-offs between flood control and power generation.

It is clear from the studies mentioned above that DCCS research is primarily focused on two concepts: first, taking forecast uncertainty into account and hence optimizing the upper storage bound of DCCS; second, utilizing normal storage as the upper limit of DCCS but using the optimization algorithm and identifying the optimal carryover storage to balance power generation and flood risk. However, these studies contribute to the technology and applications of DCCS and lack systematic physical and economic insights into the relationship between power generation and flood risk.

A critical issue for trade-offs between power generation and flood risk is quantifying the effects of inflow forecast uncertainty [19,20]. Although the forecasting technology has recently been significantly improved, the forecast uncertainty still cannot be avoided and should be considered in reservoir operations [21,22]. The forecast uncertainty might cause the release to exceed the downstream threshold and the probability of that event resulting in flood risk [23]. Risk tolerance is set to control the flood risk within a certain level, which is a chance constraint [24,25]. The reduction of a chance constraint to a deterministic equivalent yield is widely used in multipurpose reservoir operations [26]. This paper introduces the minimum flood-safety margin constraint proposed by Ding et al. [27] to consider the effects of forecast uncertainty and to meet downstream flood protection standards.

Another problem for the trade-off analysis is the explanation of what and how two objectives conflict under different scenarios of inflow. Deriving hedging rules based on the marginal utility principle is an effective method to describe the relationship between conflicting objectives; however, it relies on the analysis of the marginal utilities of the objectives. Draper and Lund [28] proved that the two-stage water-supply operation obeys the marginal utility principle, which elaborates the marginal utilities of storage and release, and derived the optimal hedging rules to obtain optimal carryover storage and to reduce future water-shortage risks. Following this principle, You and Cai [29,30] formulated future inflow uncertainty with Taylor expansions to inspect the water-supply principle. Zhao et al. [6] proved two-stage flood operations that considered forecast uncertainty following the marginal utility principle under expected floods. Ding et al. [27,31] came up with the marginal utility principle of a flood control-water conservation operation by evaluating the utility functions of carryover storage and future safety margin. For power generation and flood risk, evaluating the marginal utility of power generation is needed to explore whether the marginal utility principle can be applied. Zhao et al. [32] used the water balance, expressed the release by carrying over storage, and showed the marginal utility of power generation (MUPG) increasing with carryover storage during the dry season. To explore the MUPG over the whole year of operation, Tan et al. [33] proposed a successive iteration method to use the approximate MUPG instead of analytical functions. However, they did not evaluate the MUPG during the flood season under different scenarios of forecasted inflow, which results in different relationships with flood risk.

This study addresses these two problems in the complicated optimization problem that maximizes power generation and minimizes flood risk. Deriving the relationships between forecasted inflow and decision operations offers a better understanding of the optimal solutions. This research focused on three aspects: (1) a two-stage model based on the concept of DCCS was developed to mathematically express the actual needs of power generation and flood risk during the flood season; (2) the economic characteristics of power generation, forecasted inflow, and forecast uncertainty were considered comprehensively to applicable conditions for DCCS; (3) the OHR under different levels of forecast uncertainty and risk tolerance was derived, which revealed the trade-off between power generation and flood risk under different scenarios of forecasted inflow. This research provides an analytical framework to reveal trade-offs between power generation and

flood risk under different forecasted inflow, forecast uncertainties, and risk tolerance, which serves as a guideline for reservoir operational policy.

This paper is organized as follows: the study area is written in Section 2; the concept of DCCS, model formulation, and trade-off analysis are in Section 3; then, the case study is used to prove the theoretical derivation in the next section, and the conclusion is summarized in the last section.

## 2. Study Area

The Nierji basin in Northeast China (123° E~127.33° E, 48.42° N~51.68° N) is located on the Nen River across Heilongjiang Province and the Inner Mongolia Autonomous Region, as shown in Figure 1.

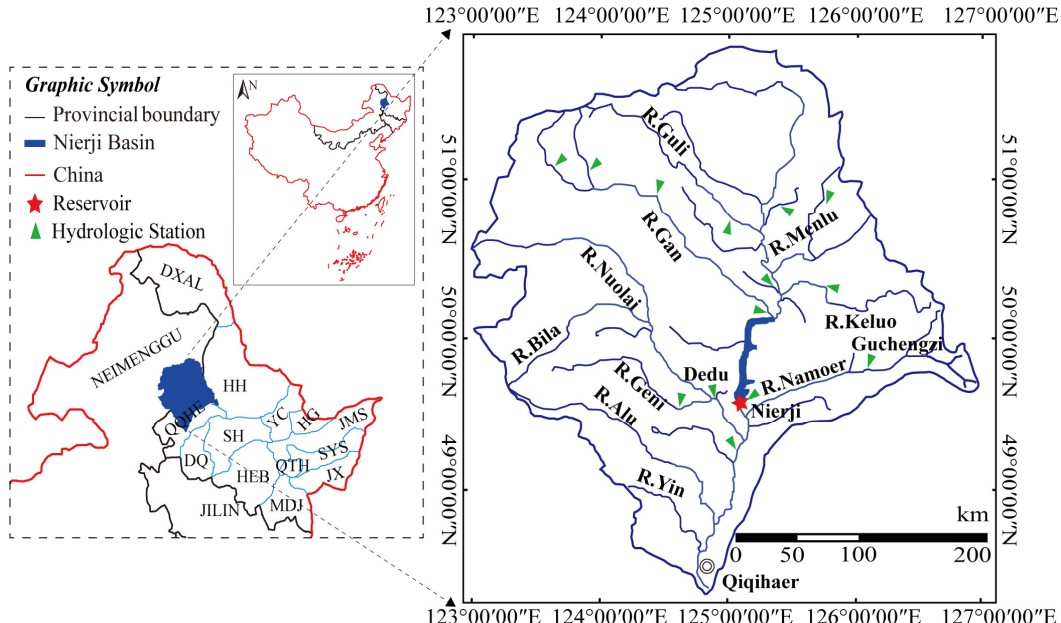

**Figure 1.** The location of the Nierji basin in China.

Nierji Reservoir is a large-scale, comprehensive water conservancy project with multi-year regulation performance that was designed to avoid flood damage in Qiqihaer City, water supply, power generation, shipping, and ecology. The normal water level of Nierji Reservoir is 216 m, the normal storage volume is $6.456 \times 10^9$ m$^3$, the flood-limited water level is 213.37 m, the flood-limited storage volume is $5.220 \times 10^8$ m$^3$, the total installed capacity is 250 MW, the total overflow capacity of the turbine unit is 1270 m$^3$/s, and the minimum downstream water demand (including industry, agriculture, urban life, ecology, etc.) is 200 m$^3$/s.

Nierji Basin has a cold-temperate continental monsoon climate, with windy and sandy conditions in the spring, warm and rainy conditions in the summer, cool conditions in the autumn, and cold and dry conditions in the winter. According to the statistics of the Nierji meteorological station, the average annual precipitation is 475 mm, decreasing from north to south, and the distribution is uneven within the year. The annual precipitation is mainly concentrated in the flood season; the total rainfall from June to September is 392.9 mm, accounting for about 70% of the total annual precipitation; winter precipitation is low, accounting for about 5% of the annual runoff; spring runoff is slightly increased, accounting for about 15% of the annual runoff; and autumn runoff accounts for about 10%.

Currently, the operation rules were designed based on two static storage volumes: (1) flood-limited storage volume ($5.220 \times 10^9$ m$^3$) during the main stage of flood season (from 21 June to 25 August) and (2) normal storage volume ($6.456 \times 10^9$ m$^3$) during the early stage of flood season (from 1 June to 20 June) and the late stage of flood season (from

26 August to 30 September). Although the operation is easy, unnecessary spills during the flood season and energy reduction in both flood and non-flood seasons may be caused. Consequently, it is necessary to increase the storage volume of the main flood season for power generation improvement without adding extra flood risk.

## 3. Methods

### 3.1. The Dynamic Control of Carryover Storage (DCCS)

With the improvement of hydrological and meteorological forecasting techniques, DCCS is operated by the related administration, implying that the storage volume can fluctuate within dynamic control bounds based on forecast information [14,16]. During DCCS, three operation modules correspond to the three stages of the flooding process, respectively [18]. The rising flood stage needs to pre-release floodwater stored above the lower bound of dynamic control. Regular flood control operations at the major flood stage are carried out in accordance with existing operating standards. Refilling at the flood recession stage can improve the floodwater storage benefits, which was the motivation of this study.

#### 3.1.1. Two-Stage Model

In practice, which module of DCCS is triggered depends heavily on the forecasted information. If there is not a large, forecasted inflow in the future, the refill operation is active and power generation is improved by utilizing the receding floodwater to increase the storage volume. If there is a large, forecasted inflow in the future, the pre-release module is activated to lower the storage volume to the flood-limited storage volume (FLSV), and then regular flood control operation is adopted to protect downstream safety from a flood event. Therefore, DCCS should consider the forecasted inflow in both the current and future stages. For the refill operation, the increased storage volume using floodwater should be reasonably determined. Though power generation grows when more floodwater is held, it may lead to spilling and increasing flood risk. On the contrary, if less floodwater is stored, the flood risk is relatively small, but power generation does not increase significantly.

To sum up, the DCCS procedure balances power generation and flood risk by considering the forecasted inflow of two stages (present and future stages) and determining the storage capacity reasonably. The decision should be revised as the forecasted inflow updates in the following operation period. As a result, the DCCS procedure follows a two-stage dynamic rolling-horizon operation scheme as shown in Figure 2. At time $t$, a rolling horizon includes $T$ periods and is divided into two stages: Stage 1 (present stage) called the decision-making stage (including one period $\Delta t$), and Stage 2 (future stage) known as the residual stage of the flood forecast horizon (including $(T-1) \cdot \Delta t$ periods), where $T$ is the forecast horizon. The initial and final storage volumes ($S_0$ and $S_2$) and the inflow forecasting in two stages (i.e., $I_1$ and $I_2$) are known in DCCS, while the release volume in two stages (i.e., $R_1$ and $R_2$) and the carryover storage ($S_1$) between the two stages are decision variables [34,35]. Only the release in Stage 1 ($R_1$) and carryover storage ($S_1$) is determined, and the window rolls over to the next period until the final scheduling period [28,29,36,37].

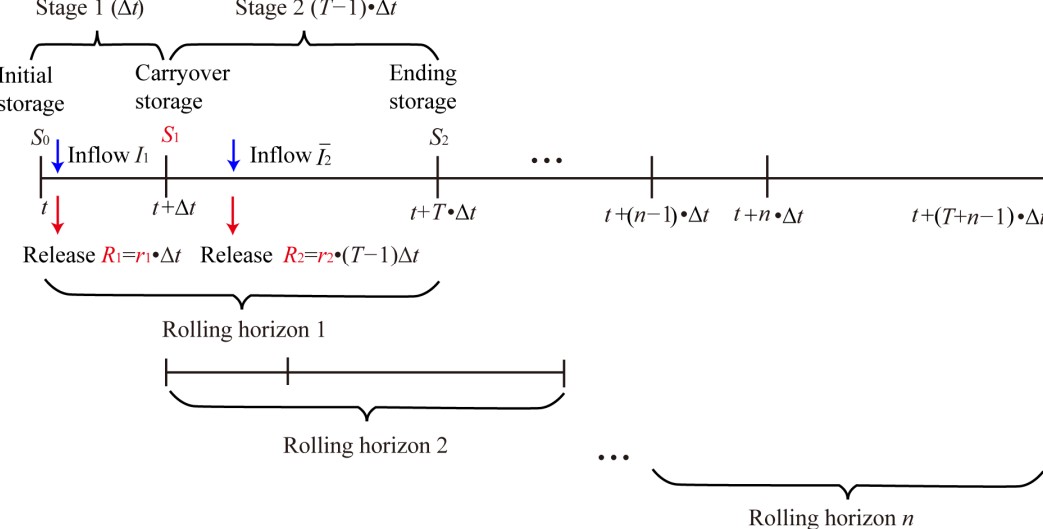

**Figure 2.** Illustration of the dynamic rolling-horizon operation scheme.

Due to the short forecast horizon and the large storage capacity, evaporation and leakage can be ignored [38], and the water-balance equations in the two stages are as follows:

$$S_0 + I_1 - r_1 \times 1 \cdot \Delta t = S_0 + I_1 - R_1 = S_1 \tag{1}$$

$$S_1 + I_2 - r_2 \times (T-1) \cdot \Delta t = S_1 + I_2 - R_2 = S_2 \tag{2}$$

where $r_k$, $I_k$, and $R_k$ are the release (in rate), inflow, and release in Stage $k$ ($k$ = 1, 2), respectively. $S_2$ is assumed to be equal to FLSV ($S^L$) to avoid additional flood risk [27]. That is, all the excess storage above the primary FLSV should be released in Stage 2, namely,

$$S_2 = S^L \tag{3}$$

It is worth mentioning that $S_2$ does not have to be equal to $S^L$ in practice, because the decision is updated when the model rolls over to the next period.

3.1.2. Forecast Errors and Flood Risk

In the real world, typical flood control systems feature an upstream reservoir for flood storage to protect the levee area downstream. In this study, flood risk represents the possibility of levee collapse caused by forecasted error rather than the dam failing [39].

Inflow forecasting error, characterized as the difference between actual and expected inflow, which includes the error of model inputs, structure, and parameters, might induce flood risk [40]. The flood risk in Stage 1 is assumed to be zero since the forecast error on a short horizon is small, and the inflow in Stage 1 is assumed to be deterministic, which is elaborated on by Ding et al. [27]. The inflow forecasting error ($\varepsilon$) and flood risk in Stage 2 are expressed as follows:

$$\varepsilon = I_2 - \bar{I}_2 = R_2 - \bar{R}_2 \tag{4}$$

$$\tau(R_2 \geq Q_{thres}) = \tau(\varepsilon \geq Q_{thres} - \bar{R}_2 = \delta) = \int_{\delta}^{+\infty} h(\varepsilon) d\varepsilon \tag{5}$$

where $I_2$ and $\bar{I}_2$ represent actual and forecasted inflows. $\bar{R}_2$ denotes the expected release in Stage 2; $\tau(\cdot)$ is the downstream flood risk, which indicates the probability of the actual release exceeding the downstream safety threshold $Q_{thres}$. $\delta$ is the flood-safety margin,

which is the difference between downstream safety threshold $Q_{thres}$ and expected release $\bar{R}_2$. $H(\cdot)$ is the probability density function of $\varepsilon$.

### 3.1.3. Power Generation in Two Stages

The power generation $E$ in two stages is the total power generation of Stage 1 ($E_1$) and Stage 2 ($E_2$),

$$E = E_1 + E_2 \tag{6}$$

The power generation at Stage $k$ ($k$ = 1, 2) $E_k$ is calculated as the product of hydro-turbine efficiency $\eta$, water head [($SSR(S_{k-1}) + SSR(S_k))/2 - SDR$], and reservoir release $R_k$ [32,40]. When the release in two stages is replaced by carryover storage according to the water balance in Equations (1) and (2), power generation $E_k$ is the function of carryover storage $S_1$, as expressed by Equation (7):

$$\begin{cases} E_1 = \eta \cdot \left[ \dfrac{SSR(S_0) + SSR(S_1)}{2} - SDR \right] \times \underbrace{(S_0 + I_1 - S_1)}_{R_1} \\[4mm] E_2 = \eta \cdot \left[ \dfrac{SSR(S_1) + SSR(S^L)}{2} - SDR \right] \times \underbrace{(S_1 + \bar{I}_2 - S^L)}_{\bar{R}_2} \end{cases} \tag{7}$$

where $SSR()$ is the stage-storage relationship of the reservoir, $SSR(S_0)$ and $SSR(S_1)$ represent the initial and ending water level of Stage 1, respectively, and $SSR(S^L)$ is the water level of $S^L$. $SDR$ is the downstream water level and can be simplified as a constant [32].

### 3.2. Model Formulation

### 3.2.1. Objective Function

A two-stage model is used to maximize power generation and minimize flood risk. In order to solve the problem of two conflicting objectives, the weighting factor $\omega$ ($0 \leq \omega \leq 1$) is employed as a compromise after normalizing the two objectives, resulting in the following objective function:

$$\begin{aligned} &\min \ G_1 + G_2 \\ &\begin{cases} G_1 = \omega \cdot (1 - \dfrac{E}{E^{\max}}) \\[4mm] G_2 = (1 - \omega) \cdot \displaystyle\int_{\delta}^{+\infty} h(\varepsilon) d\varepsilon \end{cases} \end{aligned} \tag{8}$$

where $G_1$ and $G_2$ are power generation and flood risk objectives, respectively. $E^{\max}$ is the maximum power generation in two stages without spilling and is equal to the product of installed capacity $N^{\max}$ and two stages' duration, i.e., $E^{\max} = N^{\max} \times T \cdot \Delta t$.

The marginal contributions of $S_1$ to $G_1$ and $\delta$ to $G_2$ are given by

$$\begin{cases} G_1' = -\dfrac{\omega}{E^{\max}} \cdot \dfrac{dE}{dS_1} = -\dfrac{\omega}{E^{\max}} \cdot \eta \times \dfrac{SSR'(S_1)}{2} \times (I_1 + \bar{I}_2) = -f_1(S_1) \\[4mm] G_2' = -(1 - \omega) \cdot h(\delta) = -(1 - \omega) \cdot \dfrac{1}{\sigma\sqrt{2\pi}} \exp\left\{ -\dfrac{(\delta)^2}{2\sigma^2} \right\} = -f_2(\delta) \end{cases} \tag{9}$$

$dE/dS_1$ has been proven to be positive if there is no spilled water, i.e., increasing carryover storage generates a net gain in power generation in the two-period hydropower scheduling under no spilled water [32]. $SSR'(S_1)$ is the first-order derivative of $SSR(S_1)$ and is positive, i.e., $SSR'(S_1) > 0$ [37,41]. The forecast error series ($\varepsilon$) is assumed to follow an unbiased Gaussian distribution, and the marginal contribution of $\delta$ to $G_2$ can be expressed as the second function of Equation (9) [27].

### 3.2.2. Constraints

Regular constraints for reservoir management problems, including water balance, downstream minimum water demand, storage capacity, and tolerable risk, are converted into the constraints for decision variables, i.e., $S_1$ and $\delta$, shown as follows:

$$s.t. \begin{cases} S_1 + \bar{I}_2 - (Q_{thres} - \delta) = S^L \\ S_1 \leq S_1^C = S_0 + I_1 - R_1^{\min} \\ S_1 \geq S^L \\ \delta \geq \delta^{\min} \end{cases} \tag{10}$$

where $Q_{thres} - \delta$ denotes the expected volume of release in Stage 2, i.e., $Q_{thres} - \delta = \bar{R}_2$. $S_1^C$ is the carryover storage that Stage 1 releases water at the minimum downstream water demand, i.e., $R_1 = R_1^{\min}$. It is worth noting that the downstream water demand in Stage 2 ($R_2^{\min}$) is regarded to be satisfied since all carryover storage should be released in Stage 2. The carryover storage should not be lower than $S^L$ to guarantee water supply. In the meantime, the thresholds for the tolerance of flood risk ($\tau \leq \tau_r$) should be met to guarantee downstream flood control [6,42]. That is, $\delta$ should be greater than the minimum safety margin $\delta^{\min}$, and $\delta^{\min}$ is expressed as follows when the forecast error follows an unbiased Gaussian distribution [27]:

$$\delta^{\min} = \sigma \cdot \Phi^{-1}\left(1 - \tau_r\right) \tag{11}$$

where $\Phi^{-1}(\cdot)$ is the inverse of the cumulative probability function with a standard normal distribution.

Furthermore, the carryover storage constraints for maximum power generation in two stages should be met and written in Equation (12) because they are carryover storage thresholds for their MUPG not to be zero [32]. If they are not met, there will be abandoned water, resulting in extra flood risk without power generation improvement.

$$S_1^A \leq S_1 \leq S_1^B \tag{12}$$

where $S_1^A$ stands for the minimum carryover storage and enables power generation in Stage 1 to reach its maximum value ($E_1 = E_1^{\max}$) without spilling as $dE_1/dS_1 < 0$, and $S_1^B$ is the maximum carryover storage that leads to the maximum power generation in Stage 2 without spilling ($E_2 = E_2^{\max}$) as $dE_2/dS_1 > 0$. When the initial storage capacity, inflow in two stages, the installed capacity of a reservoir $N^{\max}$, and the length of two stages are given, then $S_1^A$ and $S_1^B$ can be calculated as:

$$\begin{cases} E_1^{\max} = N^{\max} \cdot \Delta t = \eta \times \left[ \dfrac{SSR(S_0) + SSR(S_1^A)}{2} - SDR \right] \times (S_0 + I_1 - S_1^A) \\ E_2^{\max} = N^{\max} \cdot (T-1) \cdot \Delta t = \eta \times \left[ \dfrac{SSR(S_1^B) + SSR(S^L)}{2} - SDR \right] \times (S_1^B + \bar{I}_2 - S^L) \end{cases} \tag{13}$$

All these conversions are because the generation target is related to $S_1$, while the flood risk target is only related to $\delta$ according to Equation (9).

### 3.2.3. Optimal Conditions

For convex programming, the Karush Kuhn Tucker (KKT) condition is the sufficient and necessary condition [43]. The solution of the model must be the extreme value point and the global optimal when it satisfies the KKT conditions [44,45]. According to the derivation above, the objective function is concave and all constraint functions are convex; thus, the problem is a convex programming problem, and KKT conditions for the two-stage model are written as:

$$\begin{cases} G_1' + \lambda - \mu_1^{sl_1} - \mu_1^{sl_2} + \mu_1^{su_1} + \mu_1^{su_2} = 0 \\ G_2' + \lambda - \mu_2^{\delta} = 0 \\ \lambda(S_1^* + \bar{I}_2 - Q_{thres} + \delta^* - S^L) = 0 \\ \mu_1^{sl_1}(S_1^A - S_1^*) = 0 \\ \mu_1^{sl_2}(S^L - S_1^*) = 0 \\ \mu_1^{su_1}(S_1^* - S_1^C) = 0 \\ \mu_1^{su_2}(S_1^* - S_1^B) = 0 \\ \mu_2^{\delta}(\delta^{min} - \delta^*) = 0 \end{cases} \qquad (14)$$

where $\lambda$, $\mu_1^{su}$, $\mu_1^{sl}$, and $\mu_2^{\delta}$ are shadow prices that quantify the additional value resulting from changing the constraints by one unit and satisfying that $\lambda \neq 0$, $\mu_1^{su}$, $\mu_1^{sl}$, $\mu_2^{\delta} \geq 0$. The optimal KKT conditions are obtained from Equation (14):

$$G_1' - \mu_1^{sl_1} - \mu_1^{sl_2} + \mu_1^{su_1} + \mu_1^{su_2} = G_2' - \mu_2^{\delta} = -\lambda \qquad (15)$$

When all inequality constraints in Equation (14) are unbound, that is, $\mu_1^{sl_1} = \mu_1^{sl_2} = \mu_1^{su_1} = \mu_1^{su_2} = \mu_2^{\delta} = 0$, then Equation (15) transforms:

$$f_1(S_1^*) = f_2(\delta^*) = \lambda \qquad (16)$$

Equation (16) indicates that the optimal condition obeys the marginal utility principle [28].

### 3.3. Trade-Offs between Power Generation and Flood Risk

The optimal conditions of the model demonstrate that trade-offs between power generation and flood risk are allocated carryover capacity ($S_1$) and flood-safety margin ($\delta$), which is determined by the relationships between the marginal utility of power generation (MUPG, $f_1$) and that of flood risk (MUFR, $f_2$). Therefore, in this section, the MUPG under various forecasted inflow and carryover storage is evaluated to obtain the allowable inflow range for hedging and trade-offs during DCCS. Besides, the effects of forecast uncertainty and risk tolerance on trade-offs and operation decisions are analyzed.

3.3.1. Evaluation of the Marginal Utility of Power Generation

According to Equation (9), MUPG varies with inflows of two stages (i.e., $I_1$ and $\bar{I}_2$) and the carryover storage ($S_1$). To evaluate the MUPG, the three influencing factors should be discussed separately, where MUPG was proven to increase with $S_1$ by Zhao et al. [32], as shown by the solid blue line in Figure 3. The influence of $I_1$ is analyzed through two scenarios, that is, a relatively large $I_1$ resulting in $S_1^A \geq S^L$ and a small $I_1$ leading to $S_1^A < S^L$.

Then, this part focuses on the impacts of $\bar{I}_2$ under each scenario since $\bar{I}_2$ is directly engaged in the allocation of $S_1$ and $\delta$.

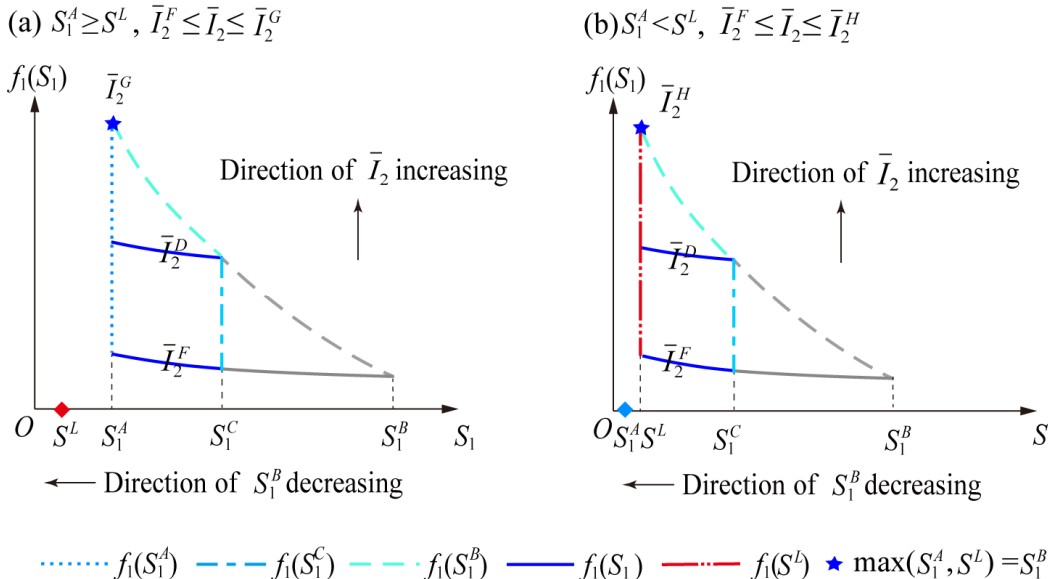

**Figure 3.** The variation of MUPG under different lower bounds as carryover storage increases when the forecasted inflow ranges from $\bar{I}_2^F$ to $\bar{I}_2^G$: (**a**) lower bound is $S_1^A$ ($S_1^A \geq S^L$); (**b**) lower bound is $S^L$ ($S_1^A < S^L$).

When the given $I_1$ is relatively large and triggers the maximum power generation in the Stage 1 constraint, i.e., $S_1^A \geq S^L$, the lower bound of carryover storage is $S_1^A$, as shown in Figure 3a. The bottom solid curve depicts the variation of MUPG with $S_1$ under the forecasting inflow $\bar{I}_2^F$, which equals downstream water demand (i.e., $\bar{I}_2^F = R_1^{\min}$) and is the minimum allowable inflow for trade-offs. The small forecasting inflow in Stage 2 causes the larger $S_1^B$, i.e., $S_1^B > S_1^C$, where the solid gray curve indicates the infeasibility domain of MUPG. When $\bar{I}_2$ increases from $\bar{I}_2^F$ to $\bar{I}_2^D$, the curve of MUPG moves upward, $S_1^B$ decreases, and the upper bound of the carryover storage is $S_1^C$. When Stage 1 releases available water to meet downstream water demand, the forecasting inflow $\bar{I}_2^D$ causes power generation in Stage 2 to peak, i.e., $S_1^B = S_1^C$. When $\bar{I}_2$ exceeds $\bar{I}_2^D$, the upper bound of carryover storage is $S_1^B$, and the MUPG curve shortens with increasing $\bar{I}_2$. When $\bar{I}_2 = \bar{I}_2^G$, power generation in two stages reaches its maximum, i.e., $S_1^B = S_1^A$, and the MUPG curve becomes a point. As a result, $\bar{I}_2^G$ is the maximum forecasting inflow allowed for trade-offs during DCCS, and the forecasting inflow in the Stage 2 within $[\bar{I}_2^F, \bar{I}_2^G]$ allows for trade-offs under relatively large $I_1$.

However, when the given $I_1$ is relatively small, leading to $S_1^A < S^L$, the lower bound of carryover storage is $S^L$, as shown in Figure 3b. $\bar{I}_2^H$ is the forecasted inflow trigger of the maximum power generation of Stage 2 and FLSV constraints at the same time (i.e., $S_1^B = S^L$). Then, the forecasting inflow in Stage 2 within $[\bar{I}_2^F, \bar{I}_2^H]$ is allowed for trade-offs under a relatively small $I_1$.

### 3.3.2. The Starting and Ending Points of Hedging

The starting and ending conditions (SWA and EWA) of hedging are important to characterize the range of hedging Figure 4 shows the SWA and EWA at the lower bound of carryover storage ($S_1^A$).

As illustrated in Figure 4, $\bar{I}_2^{SWA}$ and $\delta^{SWA}$ represent the forecasted inflow and flood-safety margin at the starting point (SWA) of hedging, causing the MUPG ($f_1(S_1^A)$) at the

lower bound of carryover storage ($S_1^A$) to equal the marginal utility of flood risk (MUFR, $f_2(\delta^{SWA})$), i.e., $f_1(S_1^A) = f_2(\delta^{SWA})$. Equation (17) can be used to calculate $\bar{I}_2^{SWA}$ and $\delta^{SWA}$.

$$
\begin{cases}
\dfrac{\omega}{E^{\max}} \cdot \eta \times \dfrac{SSR'(S_1^A)}{2} \times (I_1 + \bar{I}_2^{SWA}) = \dfrac{1-\omega}{\sigma\sqrt{2\pi}} \cdot \exp\left\{ -\dfrac{(\delta^{SWA})^2}{2\sigma^2} \right\} \\
\bar{I}_2^{SWA} = Q_{thres} + S^L - S_1^A - \delta^{SWA}
\end{cases}
\tag{17}
$$

When the lower bound is $S^L$, the $S_1^A$ in Equation (17) needs to be replaced by $S^L$. $\bar{I}_2^{SWA}$ and $\delta^{SWA}$ under the lower bound of carryover storage $S^L$ are written as follows, illustrating that $\bar{I}_2^{SWA}$ and $\delta^{SWA}$ vary with the inflow of Stage 1.

$$
\begin{cases}
\dfrac{\omega}{E^{\max}} \cdot \eta \times \dfrac{SSR'(S^L)}{2} \times (I_1 + \bar{I}_2^{SWA}) = \dfrac{1-\omega}{\sigma\sqrt{2\pi}} \cdot \exp\left\{ -\dfrac{(\delta^{SWA})^2}{2\sigma^2} \right\} \\
\bar{I}_2^{SWA} = Q_{thres} - \delta^{SWA}
\end{cases}
\tag{18}
$$

$\bar{I}_2^{EWA}$ and $\delta^{EWA}$ are the forecasted inflow and flood-safety margin at the ending point of hedging, which make the MUPG ($f_1(S_1^C)$) at the upper bound of carryover storage ($S_1^C$) equal the marginal utility of flood risk (MUFR, $f_2(\delta^{EWA})$), i.e., $f_1(S_1^C) = f_2(\delta^{EWA})$, and they are obtained by Equation (19),

$$
\begin{cases}
\dfrac{\omega}{E^{\max}} \cdot \eta \times \dfrac{SSR'(S_1^C)}{2} \times (I_1 + \bar{I}_2^{EWA}) = \dfrac{1-\omega}{\sigma\sqrt{2\pi}} \cdot \exp\left\{ -\dfrac{(\delta^{EWA})^2}{2\sigma^2} \right\} \\
\bar{I}_2^{EWA} = Q_{thres} + S^L - S_1^C - \delta^{EWA}
\end{cases}
\tag{19}
$$

However, the MUPG and MUFR cannot be equal when the upper bound of carryover is $S_1^B$ in real-world operation because of the forecast uncertainty. When $S_1 = S_1^B$, the power generation of Stage 2 has its maximum value without abandoned water. If the forecasted inflow is smaller than the actual inflow, the reservoir fails to increase power generation, which is accompanied by abandoned water and increasing flood risk. Thus, the MUFR is larger than the MUPG in this situation. SWA and EWA points, as shown in Figure 3, divide the relationship between MUPG and MUFR into three categories: R.1, R.2, and R.3.

R.1: When the forecasted inflow ($\bar{I}_2$) ranges from $\bar{I}_2^G$ to $\bar{I}_2^{SWA}$ or from $\bar{I}_2^H$ to $\bar{I}_2^{SWA}$, the relatively large inflow illustrates a large flood risk, and the MUFR exceeds MUPG under optimal conditions in Appendix B1, i.e., $f_2(\delta^*) > f_2(\delta^{SWA}) = f_1(S_1^A)$ or $f_2(\delta^*) > f_2(\delta^{SWA}) = f_1(S^L)$. As a result, carryover storage should be kept at its lowest level in order to accommodate a large, forecasted inflow and to reduce the potential flood risk in Stage 2. Under the lower bound $S_1^A$, the optimal carryover storage and flood-safety margin are $S_1^* = S_1^A$, $\delta^* = Q_{thres} + S^L - \bar{I}_2 - S_1^A$. When the lower bound is $S^L$, they are written as $S_1^* = S^L$, $\delta^* = Q_{thres} - \bar{I}_2$.

R.2: When $\bar{I}_2$ ranges from $\bar{I}_2^{SWA}$ to $\bar{I}_2^{EWA}$, more carryover storage brings more power generation but also higher flood risk. The optimal carryover storage and flood-safety margin are to equalize the marginal utility, i.e., $f_2(\delta^*) = f_1(S_1^*)$.

R.3: When $\bar{I}_2$ ranges from $\bar{I}_2^{EWA}$ to $\bar{I}_2^F$, the relatively small inflow shows that floodwater can be carried over to the next stage as much as possible, and the MUPG is always greater than MUFR based on Equation (A9), i.e., $f_1(S_1^C) > f_2(\delta^{EWA}) = f_2(\delta^*)$. Therefore, the carryover storage should be kept at the upper bound of carryover storage, namely, Stage 1 releases the downstream water demand $R$min 1, and the remaining available water from Stage 1 is carried over to Stage 2 to increase power generation due to the relatively small forecasted inflow, i.e., $S_1^* = S_1^C$, $\delta^* = Q_{thres} + S^L - \bar{I}_2 - S_1^C$.

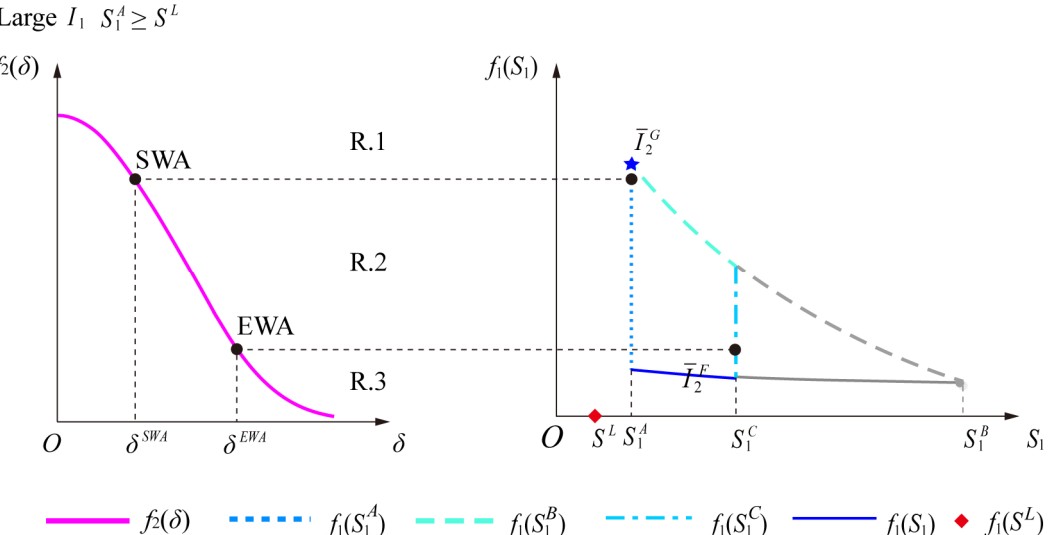

**Figure 4.** The relationship between MUFR (**left**, $f_2$) and MUPG (**right**, $f_1$) when the inflow is relatively large and the lower bound is $S_1^A$ without considering minimum flood-safety margin constraint.

### 3.3.3. Effects of Forecast Uncertainty and Risk Tolerance

The forecasted inflows and flood-safety margins at SWA and EWA (i.e., $\delta^{SWA}$ and $\delta^{EWA}$) are related to the forecast uncertainty of inflow ($\sigma$) based on Equations (17)–(19), illustrating that the hedging range (the difference between $\delta^{EWA}$ and $\delta^{SWA}$) varies with $\sigma$. However, extremely tiny or very big forecast uncertainties result in little or huge flood risk, and the specific target is power generation or flood risk. This section first derives the minimum and maximum forecast uncertainties to identify the forecast uncertainty range permitted for hedging during DCCS and then investigates the effects of forecast uncertainty and risk tolerance on the hedging range.

1.　The allowable forecast uncertainty

According to the analysis in Section 3.1, G is a special point for DCCS application under the lower bound $S_1^A$. When point G is the starting point of hedging, we have $S_1^* = S_1^A$, $\delta^G = Q_{\text{thres}} + S^L - \overline{I}_2^G - S_1^A$, $f_1(S_1^A) = f_2(\delta^G)$ (equal marginal utility principle), and the forecast uncertainty at point G is deduced from Equation (16), denoted by $\sigma^{\min}$,

$$\frac{(\delta^G)^2}{2(\sigma^{\min})^2} + Ln(\sigma^{\min}) = Ln\left( \frac{(1-\omega)\cdot\sqrt{2\pi}\cdot E^{\max}}{\omega\cdot\eta\times\dfrac{SSR'(S_1^A)}{2}\times(I_1+\overline{I}_2^G)} \right) \tag{20}$$

$\sigma^{\min}$, causing little flood risk, is the minimum forecast uncertainty for hedging, and power generation plays the dominant role within the allowable inflow range for DCCS application. There is no hedging between the two objectives until the forecasted inflow equals $\overline{I}_2^G$, and the power generation in two stages both reaches its maximum values (i.e., $S_1^A = S_1^B$, $\overline{I}_2 = \overline{I}_2^G$). According to Equation (15), the unique hedging point G is as shown in Figure 5a, and the optimal carryover storage is always equal to the upper bounds because MUPG is greater than MUFR (i.e., $f_1 > f_2$).

The other extreme case of hedging is when point F is the starting point of hedging and we have $S_1^* = S_1^A$, $\delta^F = Q_{\text{thres}} + S^L - \overline{I}_2^F - S_1^A$, $f_1(S_1^A) = f_2(\delta^F)$ (equal marginal utility principle), and the forecast uncertainty at point F is expressed as $\sigma^{\max}$,

$$\frac{(\delta^F)^2}{2(\sigma^{max})^2} + Ln(\sigma^{max}) = Ln\left( \frac{(1-\omega)\cdot\sqrt{2\pi}\cdot E^{max}}{\omega\cdot\eta\times\dfrac{SSR'(S_1^A)}{2}\times(I_1+\overline{I}_2^F)} \right) \tag{21}$$

$\sigma^{max}$ represents the maximum forecast uncertainty allowed for hedging, illustrating a relatively high flood risk, and the carryover storage remains at the lower bound (i.e., $S_1^* = S_1^A$) to reduce the flood risk. There is no hedging between two objectives until the forecasted inflow is as small as the downstream water demand ($\overline{I}_2 = \overline{I}_2^F$). Because MUPG is less than MUFR (i.e., $f_1 < f_2$), when $\overline{I}_2 > \overline{I}_2^F$, there is only one hedging point F, as shown in Figure 5b.

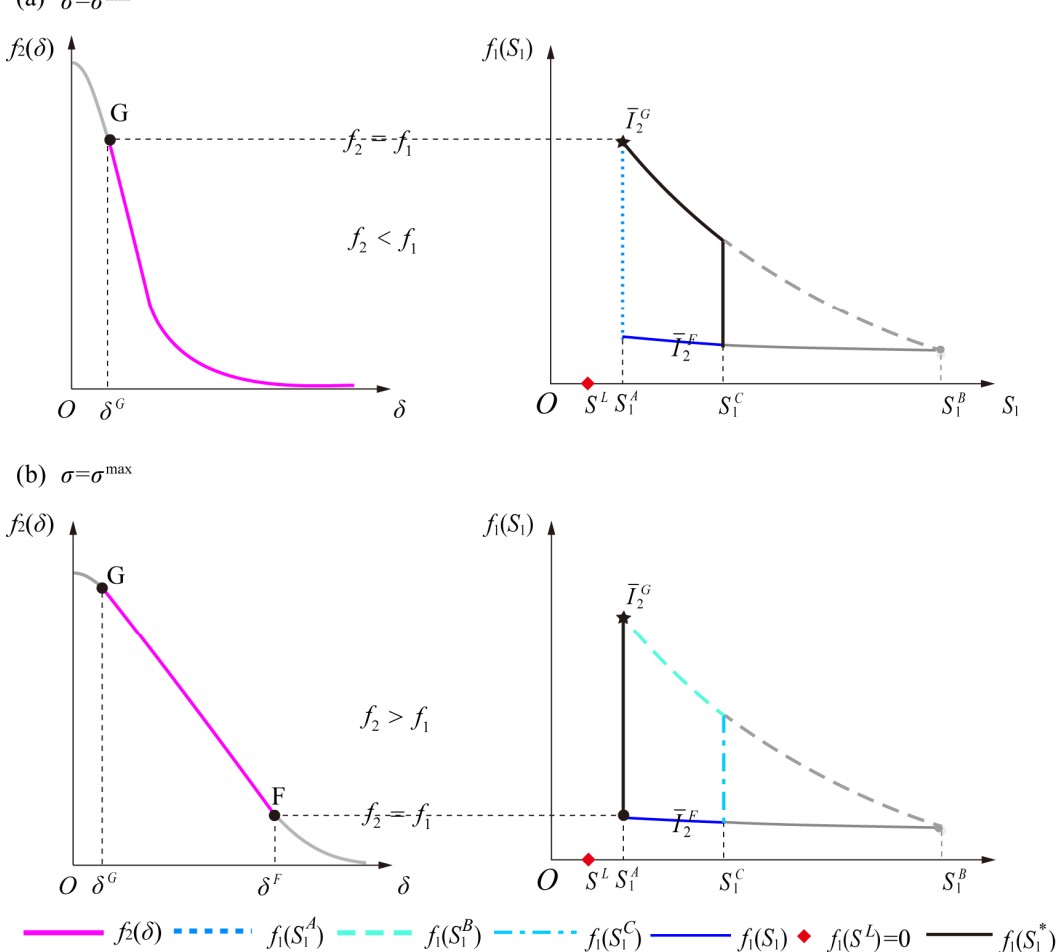

**Figure 5.** Relationship between MUFR (**left**, $f_2$) and MUPG (**right**, $f_1$) under minimum and maximum forecast uncertainties when the lower bound is $S_1^A$: (**a**) $\sigma = \sigma^{min}$; (**b**) $\sigma = \sigma^{max}$.

However, H and E are critical points for hedging when the lower bound changes to $S^L$, as shown in Figure 6, and they are similar to the points G and F, respectively. When point H is the starting point of hedging, we have $S_1^* = S^L$, $\delta^H = Q_{thres} - \overline{I}_2^H$, $f_1(S^L) = f_2(\delta^H)$, and $S_1^A$ and $\overline{I}_2^G$ in Equation (20) for solving $\sigma^{min}$ are replaced by $S^L$ and $\overline{I}_2^H$, respectively. If point E is the starting point of hedging, we have $S_1^* = S^L$, $\delta^E = Q_{thres} - \overline{I}_2^F$, $f_1(S^L) = f_2(\delta^E)$, and $S_1^A$ in Equation (21) for $\sigma^{max}$ becomes $S^L$. Equation (22) expresses $\sigma^{min}$ and $\sigma^{max}$ under the lower bound $S^L$:

$$\begin{cases} \dfrac{(\delta^H)^2}{2(\sigma^{\min})^2} + Ln(\sigma^{\min}) = Ln\left( \dfrac{(1-\omega)\cdot\sqrt{2\pi}\cdot E^{\max}}{\omega\cdot\eta\times\dfrac{SSR'(S^L)}{2}\times(I_1+\overline{I}_2^H)} \right) \\[6ex] \dfrac{(\delta^E)^2}{2(\sigma^{\max})^2} + Ln(\sigma^{\max}) = Ln\left( \dfrac{(1-\omega)\cdot\sqrt{2\pi}\cdot E^{\max}}{\omega\cdot\eta\times\dfrac{SSR'(S^L)}{2}\times(I_1+\overline{I}_2^F)} \right) \end{cases} \qquad (22)$$

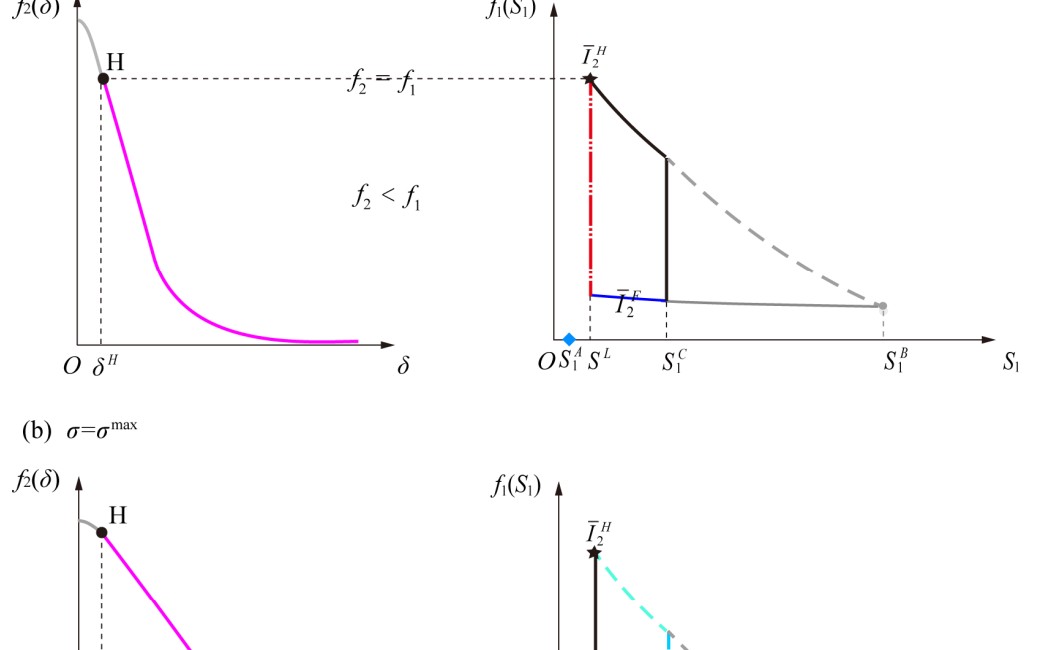

**Figure 6.** The relationship between MUFR (**left**, $f_2$) and MUPG (**right**, $f_1$) under minimum and maximum forecast uncertainties when the lower bound is $S^L$: (**a**) $\sigma = \sigma^{\min}$; (**b**) $\sigma = \sigma^{\max}$.

To summarize, trade-offs and hedging between power generation and flood risk occur only when the forecast uncertainty falls within the range defined by Equations (20)–(22), i.e., $\sigma^{\min} \leq \sigma \leq \sigma^{\max}$. If $\sigma < \sigma^{\min}$, causing too little flood risk, the power generation objective takes precedence over the flood risk objective during DCCS, which requires carryover storage as high as its upper bounds, and the optimal carryover storage under different forecasted inflows is represented by black solid curves in Figure 5a and Figure 6a, $S_1^* = \min\{S_1^B, S_1^C\}$; if $\sigma > \sigma^{\max}$, resulting in a very large flood risk, then the reduced flood risk always dominates over the increasing power generation during DCCS, which calls for carryover storage as low as its lower bounds, and optimal solutions are represented by black solid curves in Figure 5b and Figure 6b, $S_1^* = \max\{S_1^A, S^L\}$.

2.　Combined impacts of forecast uncertainty and risk tolerance

In order to assess the effects of forecast uncertainty and risk tolerance, the minimal flood-safety margin ($\delta^{min}$) was introduced by Ding et al. [23]. According to Equation (11), $\delta^{min}$ decreases as the risk tolerance ($\tau_r$) increases and the forecast uncertainty ($\sigma$) decreases. $\Delta^{SWA}$ and $\delta^{EWA}$, defined in Section 3.2, are the flood-safety margins at the start and end points of hedging, respectively, when the minimum flood-safety margin ($\delta^{min}$) constraint is unbinding. However, the flood-safety margin range for hedging may be altered when $\delta^{min}$ is considered.

Choosing a $\sigma$ from the allowable forecast uncertainty range for hedging, there are three possible cases under various $\tau_r$: Case 1 ($\delta^{min} < \delta^{SWA}$), Case 2 ($\delta^{SWA} \leq \delta^{min} \leq \delta^{EWA}$), and Case 3 ($\delta^{min} > \delta^{EWA}$), because $\tau_r$ is only related to $\delta^{min}$ but is not relevant to $\delta^{SWA}$ and $\delta^{EWA}$. The flood-safety margin range for hedging in Case 1 is the same as the unbinding $\delta^{min}$; that range is from $\delta^{min}$ to $\delta^{EWA}$ in Case 2, and there is no hedging in Case 3. That is, as risk tolerance decreases, there are fewer opportunities for hedging between power generation and flood risk, and the flood risk objective becomes more important for reservoirs.

Forecast uncertainty complicates the connections among $\delta^{min}$, $\delta^{SWA}$, and $\delta^{EWA}$ for a given $\tau_r$. To evaluate the influence of $\sigma$, the trends of increasing $\delta^{min}$, $\delta^{SWA}$, and $\delta^{EWA}$ as $\sigma$ increases are deduced in Appendix A, i.e., $d\delta^{min}/d\sigma = \Phi^{-1}(1-\tau_r) > 0$, $d\delta^{EWA}/d\sigma > d\delta^{SWA}/d\sigma > 0$, and the increase rates of $d\delta^{EWA}/d\sigma$ and $d\delta^{SWA}/d\sigma$ as $\sigma$ increase are derived to be small according to Equations (A3) and (A5). They are assumed to be zero here for simplicity (i.e., $d^2\delta^{SWA}/d\sigma^2 \approx 0$, $d^2\delta^{EWA}/d\sigma^2 \approx 0$). As a result, when the given $\tau_r$ is very small (i.e., $\Phi^{-1}(1-\tau_r) > d\delta^{EWA}/d\sigma$) or very large (i.e., $\Phi^{-1}(1-\tau_r) < d\delta^{SWA}/d\sigma$), forecast uncertainty does not influence the relationships among $\delta^{min}$, $\delta^{SWA}$, and $\delta^{EWA}$ within its allowable range [$\sigma^{min}$, $\sigma^{max}$], as illustrated in Figure 7a,b. That is, only Case 1 (i.e., $\delta^{min} < \delta^{SWA}$) exists under a large risk tolerance, whereas only Case 3 ($\delta^{min} > \delta^{EWA}$) occurs with a small risk tolerance. Furthermore, as seen in Figure 7c, increasing forecast uncertainty affects the starting of hedging when the risk tolerance is medium and satisfies $d\delta^{SWA}/d\sigma \leq \Phi^{-1}(1-\tau_r) \leq d\delta^{EWA}/d\sigma$, and the range for hedging may be narrowed.

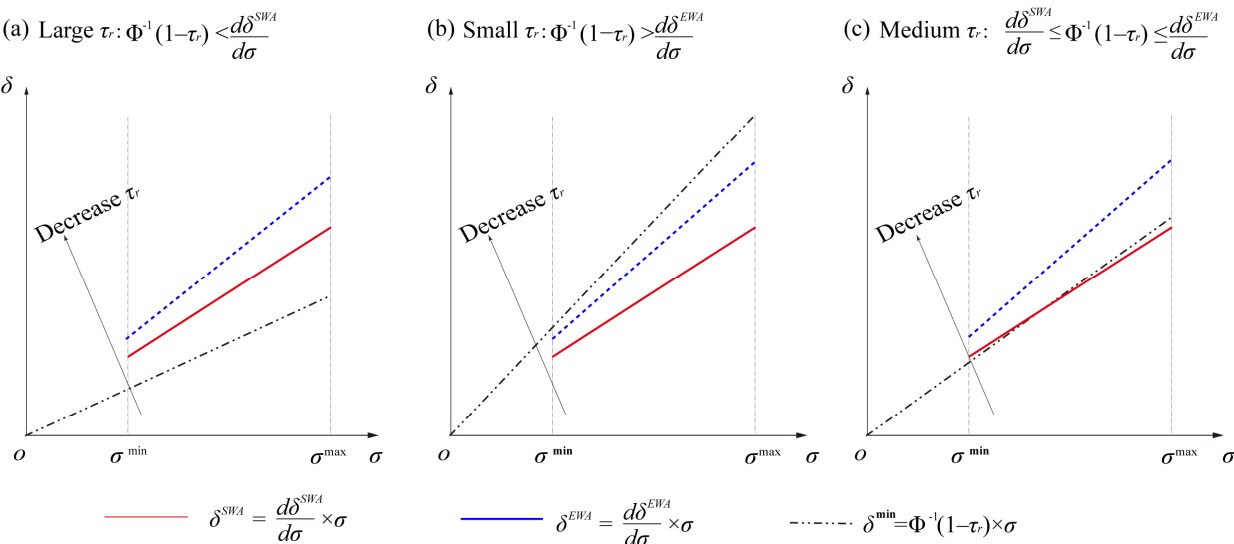

**Figure 7.** The minimum flood-safety margin ($\delta^{min}$) changes with forecast uncertainty ($\sigma$) and risk tolerance ($\tau_r$), and flood-safety margins the at start and end of hedging ($\delta^{SWA}$ and $\delta^{EWA}$) vary with different $\sigma$.

In conclusion, various risk tolerances have an independent effect on the flood-safety range for hedging, but the effects of forecast uncertainty on the hedging range are heavily dependent on the risk tolerance. However, the effects of risk tolerances and forecast uncertainty can still be generalized into three cases, namely, Case 1 ($\delta^{min} < \delta^{SWA}$), Case 2 ($\delta^{SWA} \leq \delta^{min} \leq \delta^{EWA}$), and Case 3 ($\delta^{min} > \delta^{EWA}$).

### 3.3.4. The Optimal Hedging Rules (OHR)

The optimal hedging rules (OHR) for three cases are derived based on the optimal conditions of the model and are given in Appendix B. Figure 8 and Figure 9 show the relationship between MUFR ($f_2$) and MUPG ($f_1$) for the three cases under different lower bounds of carryover storage, where the left panel shows the variation of MUFR ($f_2$) with the increasing $\delta^{min}$, and the right panel shows the variation of MUPG ($f_1$) with the decreasing $\bar{I}_2$.

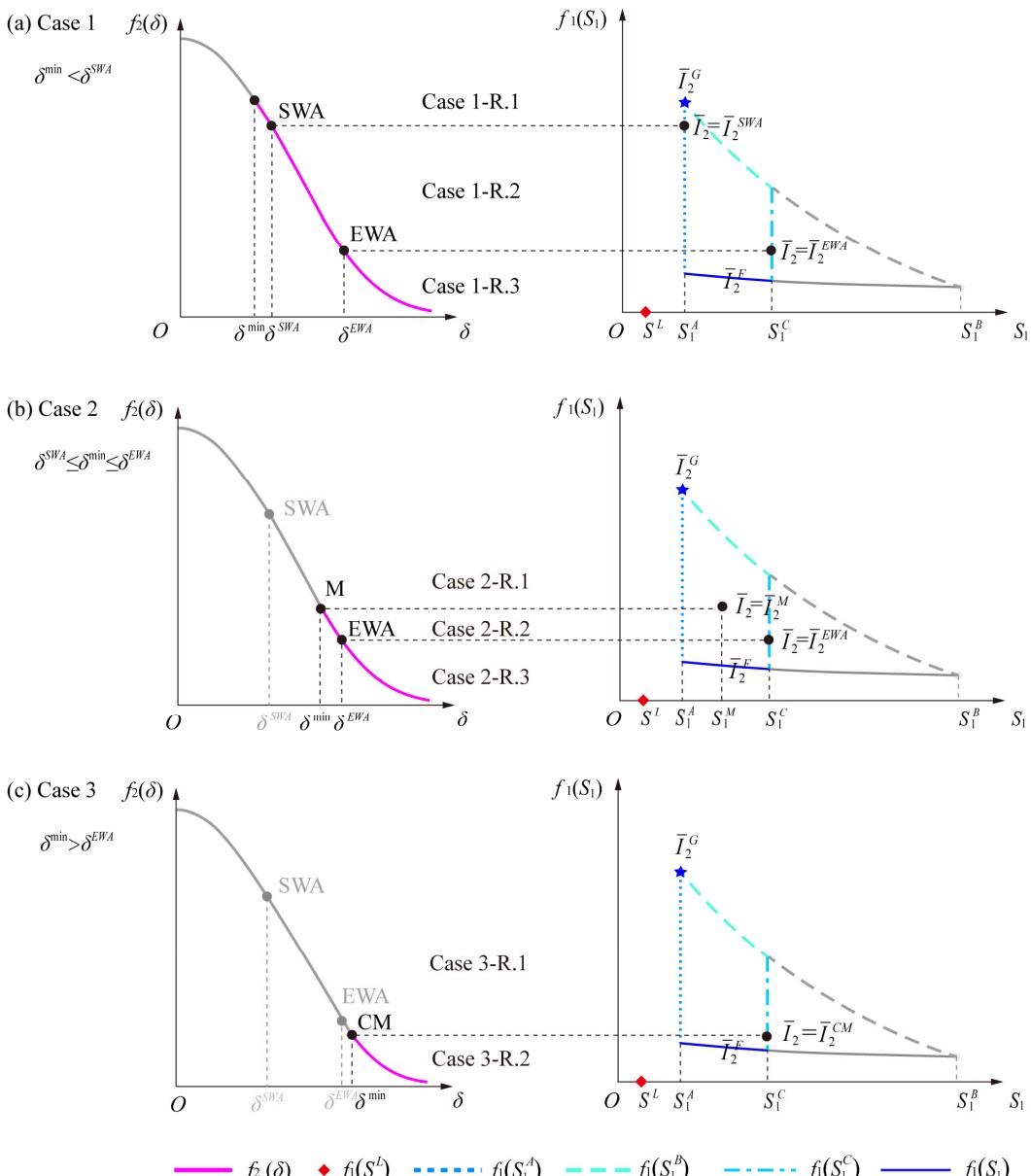

**Figure 8.** The relationship between marginal utilities of (left) flood risk ($f_2$) and (right) power generation ($f_1$) with under three values of $\delta^{min}$ when the lower bound of carryover storage is $S_1^A$: (**a**) Case 1: $\delta^{min} < \delta^{SWA}$, (**b**) Case 2: $\delta^{SWA} \leq \delta^{min} \leq \delta^{EWA}$, (**c**) Case 3: $\delta^{EWA} < \delta^{min}$.

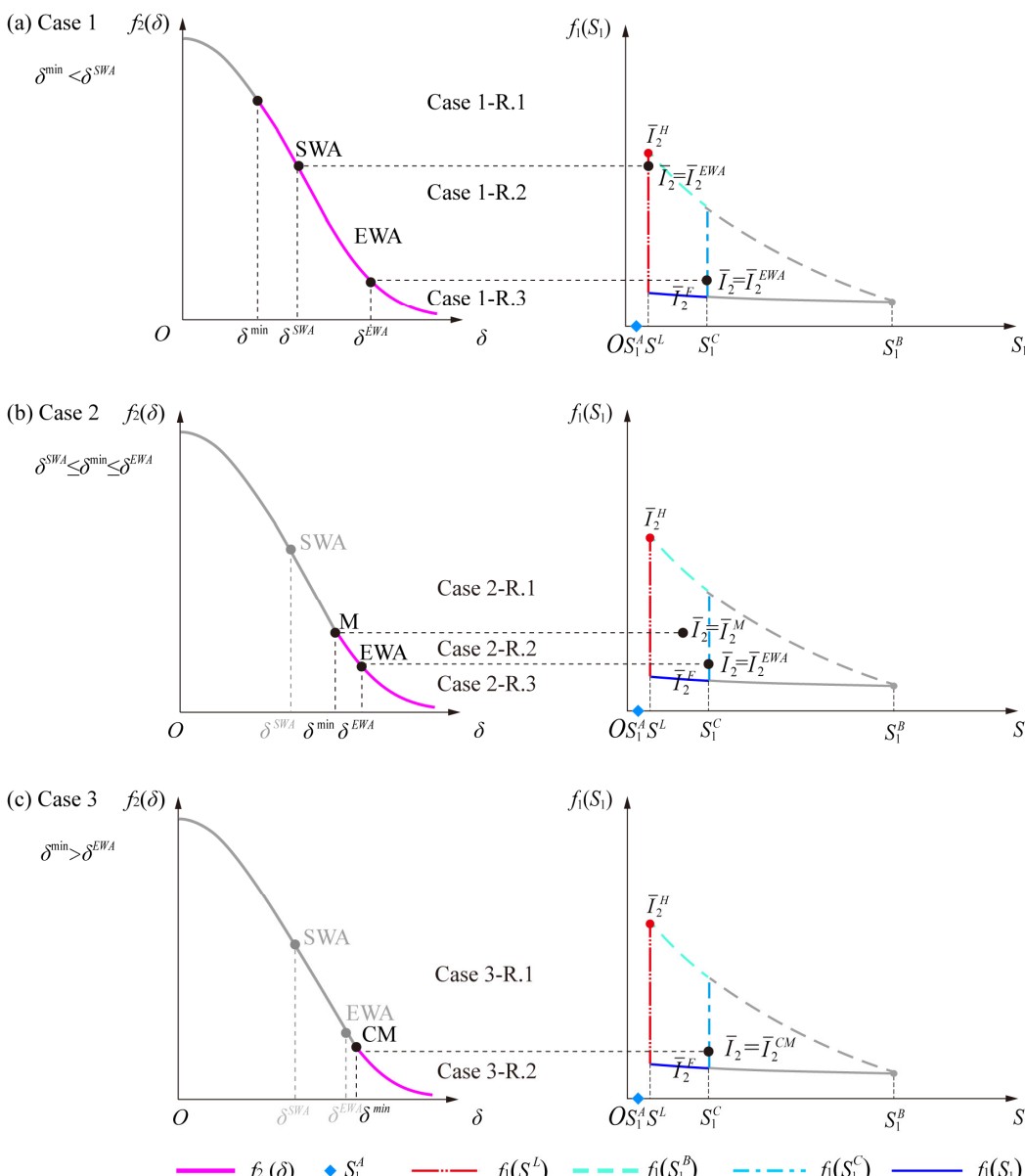

**Figure 9.** The relationship between marginal utilities of (left) flood risk ($f_2$) and (right) power generation ($f_1$) under three values of $\delta^{min}$ when the lower bound of carryover storage is $S^L$: (**a**) Case 1: $\delta^{min} < \delta^{SWA}$, (**b**) Case 2: $\delta^{SWA} \le \delta^{min} \le \delta^{EWA}$, (**c**) Case 3: $\delta^{min} > \delta^{EWA}$.

1.  Case 1: $\delta^{min} < \delta^{SWA}$

As demonstrated in Figure 8a, when the minimum flood-safety margin is smaller than the flood margin at the starting point of hedging, $\delta^{min} < \delta^{SWA}$, which has no influence on the start and end points of hedging. As a result, the optimal hedging rule under this case is the same as unbinding $\delta^{min}$. There are three relationships between MUPG and MUFR denoted as Case 1-R.1, Case 1-R.2, and Case 1-R.3 under different inflow conditions, which are the same as R.1, R.2, and R.2 in Section 3.2, respectively. The optimal solutions under different inflow conditions can be summarized as follows when the lower bound is $S_1^A$:

$$\begin{cases} S_1^* = S_1^A, \delta^* = Q_{thres} + S^L - \bar{I}_2 - S_1^A & \bar{I}_2^{SWA} \le \bar{I}_2 \le \bar{I}_2^G \\ f_1(S_1^*) = f_2(\delta^*) & \bar{I}_2^{EWA} < \bar{I}_2 < \bar{I}_2^{SWA} \\ S_1^* = S_1^C, \delta^* = Q_{thres} + S^L - \bar{I}_2 - S_1^C & \bar{I}_2^F \le \bar{I}_2 \le \bar{I}_2^{EWA} \end{cases} \quad (23)$$

When the lower bound is $S^L$, as shown in Figure 9a, the maximum allowable inflow changes from $\bar{I}_2^G$ to $\bar{I}_2^H$, and the optimal carryover storage in the first subfunction in Equation (23) is $S^L$, which is written as follows:

$$
\begin{cases}
S_1^* = S^L, \delta^* = Q_{thres} - \bar{I}_2 & \bar{I}_2^{SWA} \le \bar{I}_2 \le \bar{I}_2^H \\
f_1(S_1^*) = f_2(\delta^*) & \bar{I}_2^{EWA} < \bar{I}_2 < \bar{I}_2^{SWA} \\
S_1^* = S_1^C, \delta^* = Q_{thres} + S^L - \bar{I}_2 - S_1^C & \bar{I}_2^F \le \bar{I}_2 \le \bar{I}_2^{EWA}
\end{cases}
\tag{24}
$$

2.  Case 2: $\delta^{SWA} \le \delta^{\min} \le \delta^{EWA}$

Figure 8b and Figure 9b show the relationships between MUPG and MUFR under the medium safety margin, i.e., $\delta^{SWA} \le \delta^{\min} \le \delta^{EWA}$. Point M is the new beginning point for hedging, where the flood-safety margin is $\delta^{\min}$. The range for hedging is narrowed since $\delta^{\min}$ is greater than $\delta^{SWA}$, illustrating that much more flood-safety margin is required to reduce flood risk if the risk tolerance declines or the forecast uncertainty under a medium risk tolerance increase. The corresponding carryover storage $S_1^M$ and forecasted inflow $\bar{I}_2^M$ to point M are acquired by the marginal utility principle, i.e., $f_1(S_1^M) = f_2(\delta^{\min})$,

$$
\begin{cases}
\dfrac{\omega}{E^{\max}} \cdot \eta \times \dfrac{SSR'(S_1^M)}{2} \times (I_1 + \bar{I}_2^M) = \dfrac{1-\omega}{\sigma\sqrt{2\pi}} \cdot \exp\left\{ -\dfrac{(\delta^{\min})^2}{2\sigma^2} \right\} \\
\bar{I}_2^M = Q_{thres} + S^L - S_1^M - \delta^{\min}
\end{cases}
\tag{25}
$$

Similar to Case 1, the start and end points of hedging continue to categorize the relationship between marginal utilities of power generation and flood risk into three types, denoted by Case 2-R.1, Case 2-R.2, and Case 2-R.3.

Case 2-R.1: When the inflow in Stage 2 meets $\bar{I}_2^M < \bar{I}_2 \le \bar{I}_2^G$, according to Equation (A10), the marginal utility of power generation exceeds the maximum marginal utility of flood risk $f_2(\delta^{\min})$, i.e., $f_1(S_1) > f_1(S_1^M) = f_2(\delta^{\min})$, indicating that allocating available water after meeting the minimum safety margin requirement for carryover storage will increase power generation, i.e., $\delta^* = \delta^{\min}$, $S_1^* = Q_{thres} + S^L - \bar{I}_2 - \delta^{\min}$.

Case 2-R.2: When the forecasted inflow ranges from $\bar{I}_2^M$ to $\bar{I}_2^{EWA}$, the optimal carryover storage and flood-safety margin satisfy the marginal utility principle, i.e., $f_1(S_1^*) = f_2(\delta^*)$.

The optimal solutions for Case 2-R.3 are the same as for Case 1-R.3.

Then, the optimal solutions of Case 2 under different lower bounds can be generalized as follows:

$$
\begin{cases}
\delta^* = \delta^{\min}, S_1^* = Q_{thres} + S^L - \bar{I}_2 - \delta^{\min} & \bar{I}_2^M < \bar{I}_2 \le \bar{I}_2^G \text{ or } \bar{I}_2^M < \bar{I}_2 \le \bar{I}_2^H \\
f_1(S_1^*) = f_2(\delta^*) & \bar{I}_2^C < \bar{I}_2 < \bar{I}_2^M \\
S_1^* = S_1^C, \delta^* = Q_{thres} + S^L - \bar{I}_2 - S_1^C & \bar{I}_2^F \le \bar{I}_2 \le \bar{I}_2^C
\end{cases}
\tag{26}
$$

3.  Case 3: $\delta^{\min} > \delta^{EWA}$

As shown in Figure 8c and Figure 9c, there is no hedging between power generation and flood risk since the flood-safety margin for hedging is less than the feasible region of flood-safety margin for flood control, i.e., $\delta^* \ge \delta^{\min} > \delta^{EWA}$. $\bar{I}_2^{CM}$ is a special inflow that raises the carryover capacity to its upper bound while decreasing the optimal flood-safety margin to its lowest value, i.e., $S_1^C + \delta^{\min} = Q_{thres} + S^L - \bar{I}_2^{CM}$.

Case 3-R.1: The marginal utility of power generation always exceeds that of flood risk when the forecasted inflow ranges from $\bar{I}_2^G$ to $\bar{I}_2^{CM}$ according to the KKT conditions in Equation (A10).

Case 3-R.2: When $\bar{I}_2 < \bar{I}_2^{CM}$, the carryover storage can be kept at its upper bounds due to the small forecasted inflow, and the optimal solutions are $S_1^* = S_1^C$, $\delta^* = Q_{thres} + S^L - \bar{I}_2 - S_1^C$.

The optimal solutions under different lower bounds are the same except for the maximum allowable inflow for trade-offs and can be summarized as Equation (27):

$$
\begin{cases}
\delta^* = \delta^{\min}, \ S_1^* = Q_{thres} + S^L - \bar{I}_2 - \delta^{\min} & \bar{I}_2^{CM} < \bar{I}_2 \leq \bar{I}_2^{G} \ or \ \bar{I}_2^{CM} < \bar{I}_2 \leq \bar{I}_2^{H} \\
S_1^* = S_1^C, \ \delta^* = Q_{thres} + S^L - \bar{I}_2 - S_1^C & \bar{I}_2^{F} \leq \bar{I}_2 \leq \bar{I}_2^{CM}
\end{cases}
\tag{27}
$$

The operation operation rules under different levels of inflow forecast uncertainty and risk tolerance can be summarized in Table 1.

**Table 1.** The OHR for different inflows under various forecast uncertainty and risk tolerance.

| Cases | The Forecasted Inflow in Stage 2 for DCCS | The Optimal Solutions |
|---|---|---|
| Case 1: $\delta^{\min} < \delta^{SWA}$ | R.1: $\bar{I}_2^{SWA} < \bar{I}_2 \leq \bar{I}_2^{G}$ or $\bar{I}_2^{SWA} < \bar{I}_2 \leq \bar{I}_2^{H}$. | Little water or no water is stored, and carryover storage is kept at the lower bound of DCCS ($S_1^* = S_1^A$ or $S_1^* = S^L$). |
| | R.2: $\bar{I}_2^{EWA} < \bar{I}_2 \leq \bar{I}_2^{SWA}$. | Hydropower generation and flood risk are balanced ($f_1(S_1^*) = f_2(\delta^*)$). |
| | R.3: $\bar{I}_2^{F} < \bar{I}_2 \leq \bar{I}_2^{EWA}$. | Carryover storage volume remains at the upper bound of DCCS ($S_1^* = S_1^C$). |
| Case 2: $\delta^{SWA} \leq \delta^{\min} \leq \delta^{EWA}$ | R.1: $\bar{I}_2^{M} < \bar{I}_2 \leq \bar{I}_2^{G}$ or $\bar{I}_2^{CM} < \bar{I}_2 \leq \bar{I}_2^{H}$. | After meeting $\delta^{\min}$, the inflow from Stage 1 is carried over to Stage 2 ($S_1^* = Q_{thres} + S^L - \bar{I}_2 - \delta^{\min}$). |
| | R.2: $\bar{I}_2^{C} < \bar{I}_2 \leq \bar{I}_2^{M}$. | The optimal solutions are the same as in Case 1-R.2. |
| | R.3: $\bar{I}_2^{F} < \bar{I}_2 \leq \bar{I}_2^{C}$. | The optimal solutions are the same as in Case 1-R.3. |
| Case3: $\delta^{\min} > \delta^{EWA}$ | R.1: $\bar{I}_2^{CM} < \bar{I}_2 \leq \bar{I}_2^{G}$ or $\bar{I}_2^{CM} < \bar{I}_2 \leq \bar{I}_2^{H}$. | The optimal solutions are the same as in Case 2-R.1. |
| | R.2: $\bar{I}_2^{F} < \bar{I}_2 \leq \bar{I}_2^{CM}$. | The optimal solutions are the same as in Case 1-R.3. |

## 4. Application Results

### 4.1. Inputs of Model

In the two-stage model, we assumed that the weight $\omega$ for power generation is 0.2 and the preference for flood risk is 0.8. The reason for such an assumption is that the lower weight for power generation restricts the rise in optimal carryover storage to ensure the relatively small flood risk to guarantee downstream flood safety. However, $\omega$ is not a fixed value that can be determined by decision-makers. If decision-makers prefer to increase storage volume for improved power generation but with higher flood risk, the weight can be assigned a higher value.

*SSR(S)* is the stage-storage relationship that fits well with a function of $0.00325 \times abs(S-3913450)^{0.407} + 184.10$. The efficiency coefficient of the turbine $\eta$ equaled 8.5, and the stage downstream water level *SDR* was 184.5 m based on the Nierji Reservoir Operation Manual issued in 2014.

The forecast horizon included three days ($T = 3$), where one day was for Stage 1 (decision stage, $\Delta t = 24$ h) and two days were for Stage 2 (future stage, $(T-1)\cdot\Delta t = 48$ h). To guarantee the safety of the downstream levee, the reservoir safety discharge $Q_{thres}$ was 1600

$m^3/s$ according to the historical data. The downstream water demand, including municipal, industrial, agricultural, and environmental flow, was 200 $m^3/s$.

### 4.2. Optimal Hedging Rules Experiment

This section applies the derivation from Chapter 3 to obtain the allowable forecast uncertainty range for hedging during DCCS, as well as the range of forecast uncertainty and risk tolerance under three cases. As the carryover storage is stored at the flood recession stage, the inflow in Stage 1 should be slightly larger than that in Stage 2. To be reasonable, we used the series of inflows in stages 1 and 2 ranging from $1.036 \times 10^8$ to $1.728 \times 10^7$ $m^3$ and $1.728 \times 10^8$ to $3.456 \times 10^7$ $m^3$, respectively.

Figure 10 shows that the bounds of carryover storage vary with different inflows, in which the bounds of carryover storage ($S_1^A$, $S_1^C$) increase as the inflow in Stage 1 ($I_1$) increases, while $S_1^B$ falls as the forecasted inflow in Stage 2 ($\bar{I}_2$) rises. For $I_1 \leq 8.788 \times 10^7$ $m^3$, the lower bound is $S^L$ ($5.220 \times 10^9$ $m^3$), and the maximum power generation of Stage 1 ($S_1^A$) limits the lower bound when $I_1 > 8.788 \times 10^7$ $m^3$. The critical forecasted inflow ($\bar{I}_2^D$) for the upper bound is $1.160 \times 10^8$ $m^3$ when $I_1 = 7.540 \times 10^7$ $m^3$. That is, $S_1^C$ is the upper bound (ranging from $5.237 \times 10^9$ to $5.279 \times 10^9$ $m^3$) when $3.456 \times 10^7 \leq I_1 \leq 7.540 \times 10^7$ $m^3$ and $3.456 \times 10^7 \leq \bar{I}_2 \leq 1.160 \times 10^8$ $m^3$ because of $S_1^B \geq S_1^C$. That bound shifts to $S_1^B$, which increases from $5.279 \times 10^9$ to $5.361 \times 10^9$ $m^3$ when $7.540 \times 10^7 < I_1 \leq 9.957 \times 10^7$ $m^3$ and $1.160 \times 10^8 < \bar{I}_2 \leq 1.642 \times 10^8$ $m^3$. In particular, $S_1^A = S_1^B = 5.231 \times 10^9$ $m^3$ when $I_1 = 9.957 \times 10^7$ $m^3$ and $\bar{I}_2 = 1.642 \times 10^8$ $m^3$, illustrating that the maximum forecasted inflow for hedging ($\bar{I}_2^G$) is $1.642 \times 10^8$ $m^3$. Besides, the minimum forecasted inflow is equivalent to $3.456 \times 10^7$ $m^3$ according to the downstream water demand.

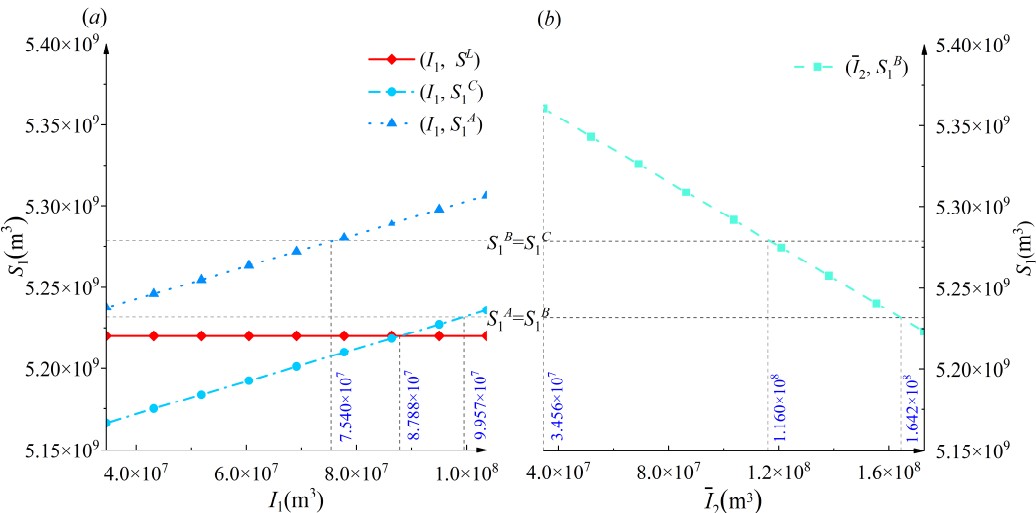

**Figure 10.** The bounds of carryover storage under different inflows: (**a**) $S_1^A$ and $S_1^C$ vary with increasing $I_1$ (**left**); (**b**) $S_1^B$ varies with increasing $\bar{I}_2$ (**right**).

According to the feasible domain of the carryover storage and the allowable inflow range obtained above, the forecast uncertainty range allowed for hedging during DCCS can be derived, using Equations (20) and (21), which is within $[2.309 \times 10^7, 5.450 \times 10^7]$.

Based on historical data, the estimated forecast error of inflows followed the normal distribution, i.e., $\varepsilon \sim N$ ($\mu = -3.11 \times 10^6$, ($\sigma = 2.368 \times 10^7)^2$). Then, the flood-safety margins at the starting and ending points of hedging were calculated via Equations (17)–(19), i.e., $\delta^{SWA} = 1.037 \times 10^8$ m$^3$ and $\delta^{EWA} = 1.054 \times 10^8$ m$^3$. The ranges of risk tolerance $\tau_r$ corresponding to three cases (i.e., $\delta^{min} < \delta^{SWA}$, $\delta^{SWA} \leq \delta^{min} \leq \delta^{EWA}$, and $\delta^{min} > \delta^{EWA}$) were derived by $\delta^{min} = \mu + \sigma \cdot \Phi^{-1}(1 - \tau_r)$ and are listed in Table 2.

Furthermore, three different risk tolerances ($5 \times 10^{-3}$, $4.50 \times 10^{-6}$, $7.93 \times 10^{-7}$) were chosen from its range obtained above to derive the forecast uncertainty ranges corresponding to three cases, and the results are shown in Table 2. When $\tau_r = 5 \times 10^{-3}$, $\delta^{min}$ is always smaller than $\delta^{SWA}$ at any forecast uncertainty within its allowable range for hedging. For example, if $\mu = 0$, $\sigma = 3.456 \times 10^7$, then $\delta^{min} < \delta^{SWA}$ can be known since $\delta^{min} = 9.210 \times 10^7$ m$^3$ and $\delta^{SWA} = 1.501 \times 10^8$ m$^3$. When $\tau_r = 4.50 \times 10^{-6}$, Case 2 ($\delta^{SWA} \leq \delta^{min} \leq \delta^{EWA}$) occurs under forecast uncertainty ranging from $2.309 \times 10^7$ to $3.456 \times 10^7$ or within $[3.974 \times 10^7, 5.450 \times 10^7]$, while Case 3 ($\delta^{min} > \delta^{EWA}$) happens at a relatively small range of forecast uncertainty, i.e., ($3.456 \times 10^7$, $3.974 \times 10^7$). When $\tau_r$ is $7.93 \times 10^{-7}$, there is no hedging between power generation and flood risk within the allowable forecast uncertainty range.

In summary, all three cases exist when risk tolerance decreases under a given forecast uncertainty and all three cases exist. However, whether one or two of the three cases exist depends on the values of the given risk tolerance, which is consistent with the theoretical derivation.

**Table 2.** The range of forecast uncertainty and risk tolerance corresponds to three cases.

| Cases | Range of $\tau_r$ $\sigma = 2.368 \times 10^7$ | Range of $\sigma$ (×10⁷) | | |
| --- | --- | --- | --- | --- |
| | | $\tau_r = 5 \times 10^{-3}$ | $\tau_r = 4.50 \times 10^{-6}$ | $\tau_r = 7.93 \times 10^{-7}$ |
| Case 1: $\delta^{min} < \delta^{SWA}$ | ($5.410 \times 10^{-6}$, 0.500) | [2.309, 5.450] | -- | -- |
| Case 2: $\delta^{SWA} \leq \delta^{min} \leq \delta^{EWA}$ | [$3.730 \times 10^{-6}$, $5.410 \times 10^{-6}$] | -- | [2.309, 3.456], [3.974, 5.450] | -- |
| Case 3: $\delta^{min} > \delta^{EWA}$ | (0.000, $5.410 \times 10^{-6}$) | -- | (3.456, 3.974) | [2.309, 5.450] |

Following that, the optimal solutions for three cases are calculated to quantify the effects of various minimum flood-safety margins on the operation decisions. The solid blue curves, red dotted curves, and black dashed curves in Figure 11 represent $\tau_r = 5 \times 10^{-3}$, $4.50 \times 10^{-6}$, and $7.93 \times 10^{-7}$, corresponding to Case 1, 2, and 3, respectively, and $\delta^{min}$ takes the values of $6.000 \times 10^7$ m$^3$, $1.047 \times 10^8$ m$^3$, $1.082 \times 10^8$ m$^3$ for the three cases under current forecast level ($\sigma = 2.368 \times 10^7$). As $\tau_r$ decreases in Figure 11, the hedging range is narrowed down, and there is no hedging when $\tau_r$ is small enough.

As shown in Figure 11a, the optimal carryover storage ($S_1^*$) for three cases declines slightly with the increasing $\delta^{min}$ under the same $\bar{I}_2$ within [$1.613 \times 10^8$ m$^3$ to $1.401 \times 10^8$ m$^3$], where they represent the inflows at the starting points of hedging for cases 1 and 2, respectively, namely, $\bar{I}_2^{SWA} = 1.613 \times 10^8$ m$^3$ and $\bar{I}_2^M = 1.401 \times 10^8$ m$^3$. For example, when $\bar{I}_2 = 1.613 \times 10^8$ m$^3$, $S_1^*$ corresponds to three cases: $5.229 \times 10^9$ m$^3$, $5.228 \times 10^9$ m$^3$, and $5.224 \times 10^9$ m$^3$, respectively. Under the same $\bar{I}_2$ ranging from $1.401 \times 10^8$ m$^3$ ($\bar{I}_2^M$) to $1.140 \times 10^8$ m$^3$ ($\bar{I}_2^{EWA}$), $S_1^*$ is the same for Cases 1 and 2 because hedging exists within the inflow range, and $S_1^*$ for Case 3 is less than for Cases 1 and 2. When $\bar{I}_2$ ranges from $1.140 \times 10^8$ m$^3$ ($\bar{I}_2^{EWA}$) to $1.122 \times 10^8$ m$^3$ ($\bar{I}_2^{CM}$), $S_1^*$ equals the upper bound of carryover storage for Cases 1 and 2, while $S_1^*$

for Case 3 is the carryover storage net of $\delta^{min}$ and less than its upper bound. If the forecasted inflow is within [$3.456 \times 10^7\,m^3$, $1.122 \times 10^8\,m^3$], then $3.456 \times 10^7\,m^3$, representing the minimum allowable inflow for hedging, i.e., $\bar{I}_2^F$, the three curves overlapping indicate that $S_1^*$ for three cases is equal. The tendency of optimal flood-safety margins ($\delta^*$) in three cases is the opposite to that of carryover storage, as shown in Figure 11b.

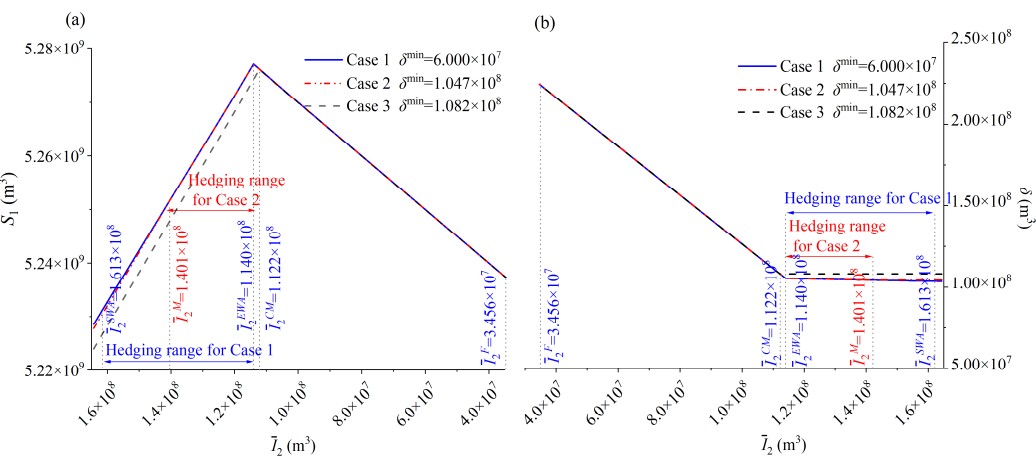

**Figure 11.** The OHR for three cases caused by three risk tolerances $\tau_r$ when confronting different inflows: (**a**) the optimal carryover storage ($S_1^*$) and (**b**) the optimal flood-safety margin ($\delta^*$).

### 4.3. Comparison with the Current Operation Rules

In this section, the OHR is used in continuous scheduling from 1981 to 2010, based on the forecasted inflow data from historical records. The operation period in flood and non-flood seasons lasts three days and ten days, respectively. Nierji's risk tolerance is 0.005 and $\delta^{min} = 6.023 \times 10^7$, thus, the optimal solutions are the same as in Case 1. The performance of OHR is compared to that of current operating rules (COR) in flood risk and power generation, as shown in Figure 12 and Table 3, respectively.

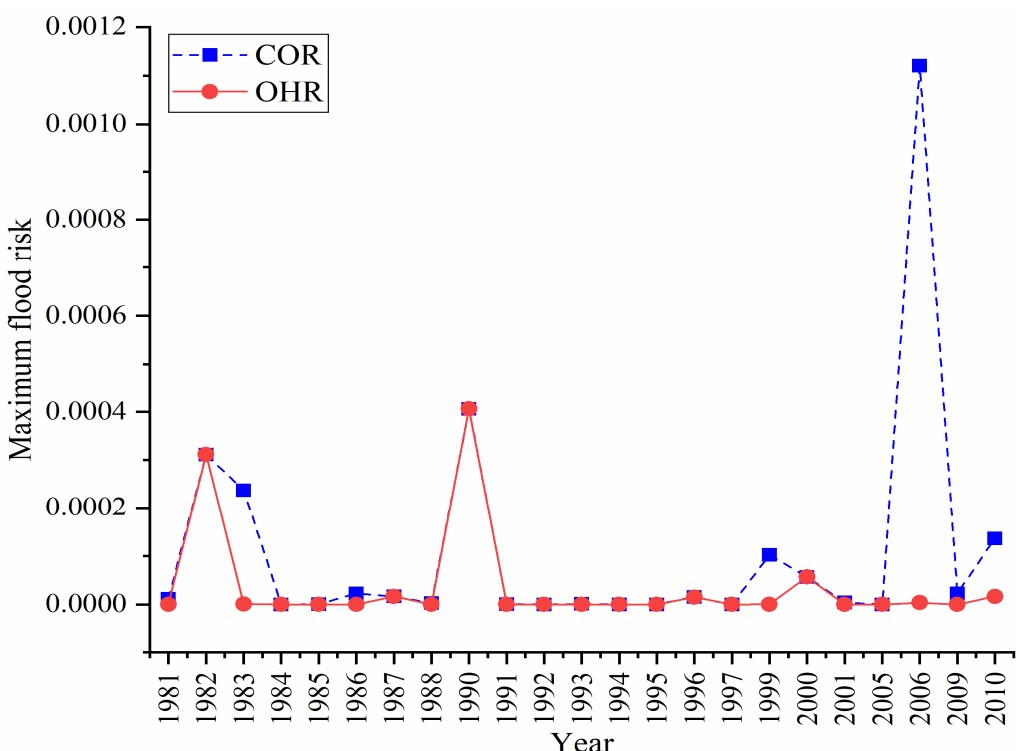

**Figure 12.** The maximum flood risk using COR and OHR from 1981 to 2010.

Figure 12 illustrates the maximum flood risk under two operation rules from 1981 to 2010. The maximum flood risk in each year for OHR is the maximum value chosen from the flood risk sets when OHR was applied, compared with the maximum flood risk selected from the flood risk sets when COR was used at the same period. The results show that the maximum flood risk can be reduced during OHR application, especially in 1981, 1983, 1986, 1999, 2006, 2009, and 2010. In fact, the maximum risk of other years was also lower, but the risk reductions are difficult to show in the graph because both the flood risk and the reduction were very small and close to 0. In addition, the maximum flood risk in extra wet years (1989 and 1998) and extra dry years (2002, 2003, 2007, and 2008) was not estimated since OHR was inapplicable. For extra wet years, the excessive inflows resulted in spills, and controlling flood risk was the unique objective, whereas in extra dry years, the inflow was insufficient to fulfill downstream water demand, and the storage volume was lower than the FLSV to satisfy downstream water demand.

**Table 3.** Comparison of the power generation between COR and OHR ($10^8$ kWh).

| Methods | Annual Average | Flood Season | Non-Flood Season | Wet Year | Normal Year | Dry Year |
|---------|----------------|--------------|------------------|----------|-------------|----------|
| COR | 6.36 | 3.23 | 3.13 | 7.93 | 6.57 | 4.63 |
| OHR | 6.62 | 3.34 | 3.28 | 8.15 | 6.74 | 4.99 |
| Change | 0.26 | 0.11 | 0.15 | 0.22 | 0.17 | 0.33 |
| Rate | 4.09% | 4.02% | 4.79% | 2.77% | 2.59% | 7.13% |

Table 3 shows that, in comparison to the COR, the OHR outperformed in terms of increasing power generation on an annual average in flood and non-flood seasons, as well as in wet, normal, and dry years. Although both flood and non-flood seasons benefitted from the elevated storage volume for improving power generation, the non-flood season saw more increases than the flood season. For wet, normal, and dry years, the OHR had the best performance in dry years.

Furthermore, the scheduling processes for three hydrological years were demonstrated in order to investigate storage volume under OHR, with 1985, 1994, and 2005 representing wet, normal, and dry years, respectively. As shown in Figure 13, the storage volume increased from 1 July to 11 August and remained at $5.220 \times 10^9$ m³ at the end of the main flood season in 1985, despite being at the flood recession stage, because the forecasted inflow at the end of the main flood season continued to exceed its maximum allowable inflow for DCCS. The storage volume in 1994 and 2005 was raised using OHR almost the whole main flood season, but the raised storage volume at the end of the main flood season in 1994 was greater than that in 2005 ($5.282 \times 10^9$ m³ versus $5.265 \times 10^9$ m³) since the inflow at the end of the main flood season in 2005 was less than the downstream water demand. Over these three years, OHR performance was beneficial in increasing storage capacity and limiting energy losses due to inflow constraints.

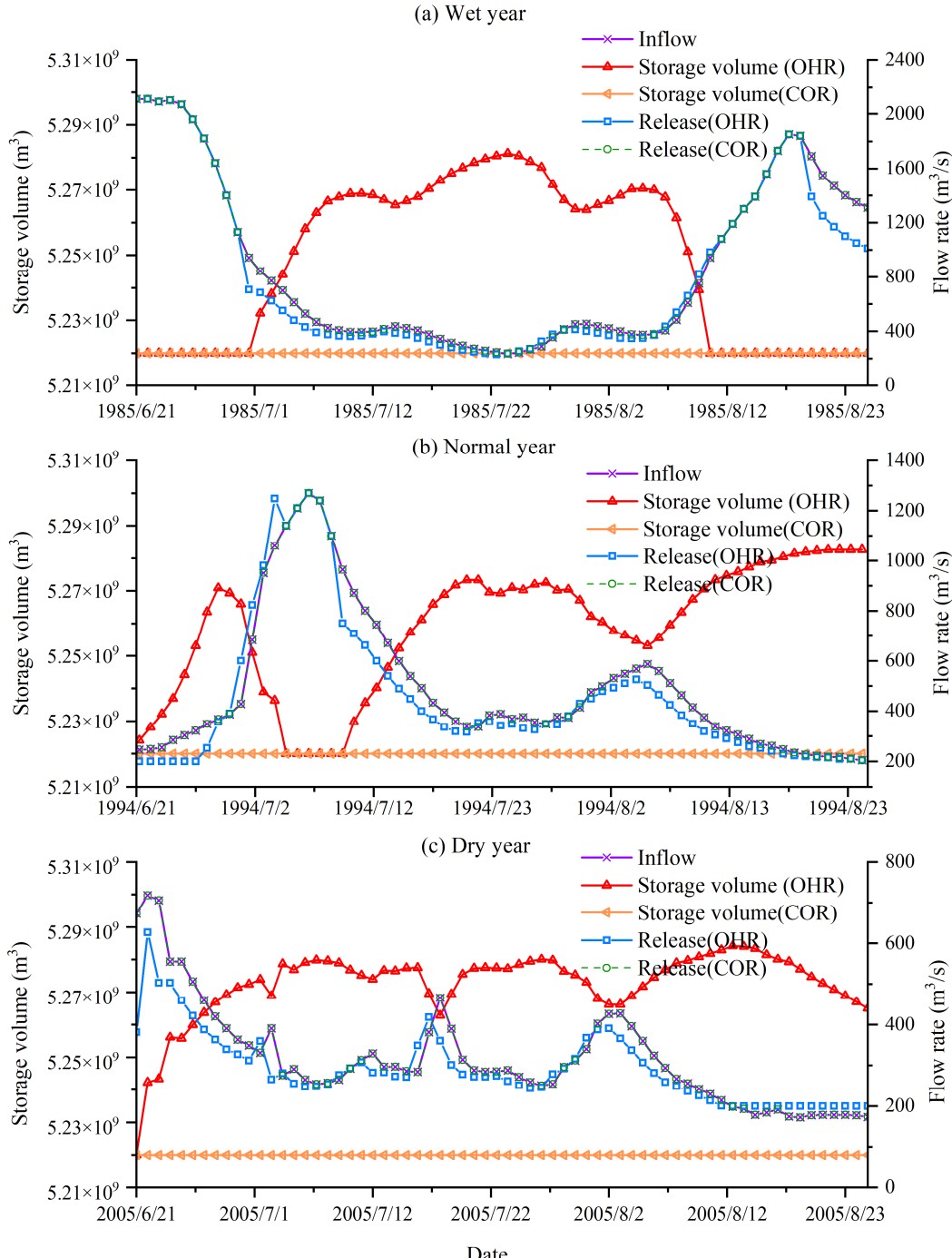

**Figure 13.** Comparison of storage volume for OHR and COR under a: (**a**) wet year; (**b**) normal year; (**c**) dry year.

## 5. Conclusions

This study focused on hydro-economic and mathematical analysis and derived the OHR to reveal trade-offs between power generation and flood risk under different forecasted inflows, forecast uncertainties, and risk tolerance. The conclusions are summarized as follows:

(1)  Hedging and trade-offs between power generation and flood risk exist during DCCS only when the forecasted inflow is greater than the minimum downstream water demand and less than the inflow that allows power generation in two stages to reach its peak without spilling.

(2)  We identified the forecast uncertainty range that allows for hedging between two objectives by calculating the minimum and maximum forecast uncertainties. If the forecast uncertainty is greater than its maximum, reducing flood risk is the unique objective considered by decision-makers. If the forecast uncertainty is less than its minimum, the carryover storage keeping at its upper bounds meets the current flood control standard.

(3)  Compared to forecast uncertainty, downstream risk tolerance plays a more important role in determining which case of the OHR is adopted in real-world operations.

(4)  In the real-world application, compared with COR, OHR had an excellent performance in power generation improvement in dry years, indicating that OHR can alleviate the energy crisis during dry years.

It should be noted that this method is suitable for areas with less abundant water resources, and extra wet or extra dry regions cannot be applied. Furthermore, some simplifications and assumptions, such as the given weight and forecast errors following a Gaussian distribution and the adaptation to climate change, should be studied in the future.

**Author Contributions:** Conceptualization, L.Z., W.D. and Y.P.; methodology, L.Z.; software, L.Z., W.D. and S.J.; validation, W.D., curation, L.Z.; writing—original draft preparation, L.Z.; writing—review and editing, J.R.L., W.D. and X.Z.; supervision, G.W. All authors have read and agreed to the published version of the manuscript.

**Funding:** This work was supported by the National Key Research and Development Project of China (2016YFC0400903), the National Natural Science Foundation of China (Grants No. 51779030, 51709108, 52079015, 51925902, 51709036).

**Data Availability Statement:** Not applicable.

**Acknowledgments:** The first author would like to acknowledge the Chinese Scholarship Council (CSC) for supporting her Ph.D. study at the University of California, Davis (UCD).

**Conflicts of Interest:** The authors declare that they have no conflict of interest.

## Nomenclature

| | |
|---|---|
| OHR | The optimal hedging rules. |
| DCCS | Dynamic control of carryover storage. |
| FLSV($S^L$) | Flood-limited storage volume. |
| $\Delta t$ | One period of two-stage operation. |
| $T$ | Forecast horizon. |
| $S_0$, $S_k$ | Initial storage volume of Stage 1 and storage at the end of Stage $k$ ($k$ = 1, 2). |
| $I_k$, $R_k$ | Actual inflow and release volume in Stage $k$ ($k$ = 1, 2). |
| $\bar{I_2}$, $\bar{R_2}$ | Forecasted inflow and release volume in Stage 2. |
| $\varepsilon$ | Inflow forecasting error in Stage 2. |

| $Q_{thres}$ | The threshold discharge capacity for the downstream safety. |
|---|---|
| $\delta$ | Forecasted flood-safety margin. |
| $\sigma$ | Forecast uncertainty of inflow in Stage 2. |
| $E_k$ | Power generation in stage $k$ ($k$ = 1, 2). |
| $SSR(S_0)$, | The initial stage-storage relationship water level in Stage 1 and stage-storage |
| $SSR(S_k)$ | relationship at the end of Stage $k$ ($k$ = 1, 2). |
| $SDR$ | Downstream water level. |
| $G_1$, $G_2$ | Power generation and flood risk objectives, respectively. |
| $G_1^{'}$, $G_2^{'}$ | Marginal utilities of power generation and flood risk, respectively. |
| $\omega$ | The weight is designed for power generation. |
| $\delta^{min}$ | Minimum flood-safety margin required for flood risk in Stage 2. |
| $E_k^{max}$ | Maximum power capacity in Stage $k$ ($k$ = 1, 2). |
| $S_1^A$ | The lower bound of carryover storage originated from maximum power generation of Stage 1. |
| $S_1^B$ | The upper bound of carryover storage originated from maximum power generation of Stage 2. |
| $R_1^{min}$ | Minimum downstream water demand. |
| $S_1^C$ | The upper bound of carryover storage originated from downstream water demand of Stage 2. |
| $\tau_r$ | Risk tolerance. |
| $\bar{I}_2^G$ | A specific forecasted inflow that triggers the maximum power generation in two stages without spilling when the lower bound of carryover storage is $S_1^A$. |
| $\bar{I}_2^H$ | A specific forecasted inflow that triggers the maximum power generation in two stages without spilling when the lower bound of carryover storage is $S^L$. |
| $\bar{I}_2^D$ | A specific forecasted inflow that triggers the maximum power generation in Stage 2 and downstream water demand constraints at the same time. |
| $\bar{I}_2^F$ | The minimum allowable inflow for DCCS application, which is equal to the minimum downstream water demand. |
| $\delta^{SWA}$, $\bar{I}_2^{SWA}$ | Flood-safety margin and forecasted inflow at the starting hedging point that marginal utility of power generation equals marginal utility of flood risk, i.e., $f_1(S_1^A) = f_2(\delta^{SWA})$. |
| $\delta^{EWA}$, $\bar{I}_2^{EWA}$ | Flood-safety margin and forecasted inflow at the ending hedging point that marginal utility of flood risk equals marginal utility of power generation ($S_1^C$), i.e., $f_1(S_1^C) = f_2(\delta^{EWA})$. |
| $MUPG(f_1(S_1))$ | Marginal utility of power generation in Stage 1. |
| $MUFR(f_2(\delta))$ | Marginal utility of flood risk in Stage 2. |
| $\sigma^{min}$, $\sigma^{max}$ | Minimum and maximum allowed forecast uncertainty for hedging, respectively. |
| $d\delta^{SWA}/d\sigma$ | The trend of $\delta^{SWA}$ as $\sigma$ increases. |
| $d\delta^{EWA}/d\sigma$ | The trend of $\delta^{EWA}$ as $\sigma$ increases. |
| $d^2\delta^{SWA}/d\sigma^2$ | The trend of $d\delta^{SWA}/d\sigma$ as $\sigma$ increases. |
| $d^2\delta^{EWA}/d\sigma^2$ | The trend of $d\delta^{EWA}/d\sigma$ as $\sigma$ increases. |

| $S_1^*$ | Optimal carryover storage from Stage 1 to Stage 2. |
|---|---|
| $\delta^*$ | Optimal flood-safety margin in Stage 2. |
| $S_1^M$ | The carryover storage makes the marginal utility of power generation equal minimum marginal utility of flood risk, i.e., $f_2(\delta^{min}) = f_1(S_1^M)$. |
| $\bar{I}_2^M$ | The inflow for situation $f_2(\delta^{min}) = f_1(S_1^M)$. |
| $\bar{I}_2^{CM}$ | The inflow for situation $\delta = \delta^{min}$, $S_1 = S_1^C$. |

## Appendix A

According to the definition of $\delta^{min}$, the trend of $\delta^{min}$ as $\sigma$ increases ($d\delta^{min}/d\sigma$) can be expressed as:

$$\frac{d\delta^{min}}{d\sigma} = \Phi^{-1}(1 - \tau_r) \tag{A1}$$

In Equation (A1), $d\delta^{min}/d\sigma$ is positive and monotonically increases because $\Phi^{-1}(1 - \tau_r)$ is positive.

$d\delta^{SWA}/d\sigma$ illustrates the trend of $\delta^{SWA}$ as $\sigma$ increases:

$$\frac{d\delta^{SWA}}{d\sigma} = \frac{\dfrac{\left(\delta^{SWA}\right)^2}{\sigma^3} - \dfrac{1}{\sigma}}{\left(\dfrac{\delta^{SWA}}{\sigma^2} - \dfrac{1}{I_1 + \bar{I}_2^{SWA}}\right)} > \frac{\delta^{SWA}}{\sigma} - \frac{\sigma}{\delta^{SWA}} \tag{A2}$$

where the molecular $\delta^{SWA}/\sigma^2 - 1/(I_1 + \bar{I}_2^{SWA})$ is smaller than $\delta^{SWA}/\sigma^2$; therefore, $d\delta^{SWA}/d\sigma$ is greater than $\delta^{SWA}/\sigma - \sigma/\delta^{SWA}$. $\delta^{SWA}$ is defined as the difference between the threshold for downstream levees $Q_{thres}$ and the release at point SWA (i.e., $\delta^{SWA} = Q_{thres} - R_2^{SWA}$), which is larger than the difference between $Q_{thres}$ and the threshold of turbine release capacity ($R_2^{max}$) since the maximum power generation binding lead to $R_2^{SWA} \leq R^{max}$. To ensure downstream safety, the threshold of turbine release capacity was designed far lower than the downstream levee threshold [19], i.e., $Q_{thres} - R_2^{max} > \sigma$. As a result, $\delta^{SWA} = Q_{thres} - R_2^{SWA} > \sigma$ and $d\delta^{SWA}/d\sigma > \delta^{SWA}/\sigma - \sigma/\delta^{SWA} > 0$.

$d^2\delta^{SWA}/d\sigma^2$ illustrates the trend of $d\delta^{SWA}/d\sigma$ as $\sigma$ increases:

$$\frac{d^2\delta^{SWA}}{d\sigma^2} = \frac{\dfrac{2\delta^{SWA} \cdot \left(\dfrac{d\delta^{SWA}}{d\sigma}\right) \cdot \sigma^3 - 3\left(\delta^{SWA}\right)^2 \sigma^2 + \sigma^4}{(\sigma^3)^2}}{\left(\dfrac{\delta^{SWA}}{\sigma^2} - \dfrac{1}{I_1 + \bar{I}_2^{SWA}}\right)}$$
$$- \frac{\left(\dfrac{\left(\delta^{SWA}\right)^2}{\sigma^3} - \dfrac{1}{\sigma}\right) \times \dfrac{\left(\dfrac{d\delta^{SWA}}{d\sigma}\right) \cdot \sigma^2 - 2\sigma \cdot \delta^{SWA}}{(\sigma^2)^2}}{\left(\dfrac{\delta^{SWA}}{\sigma^2} - \dfrac{1}{I_1 + \bar{I}_2^{SWA}}\right)^2} \tag{A3}$$

To judge the characteristic of $d^2\delta^{SWA}/d\sigma^2$ for simplicity, assume that $\delta^{SWA} = \sigma$ and $1/(I_1 + \bar{I}_2^{SWA}) = 0$. Then, the actual value of $d^2\delta^{SWA}/d\sigma^2$ is close to $(-2/\sigma)$ and can be derived since $\sigma < \delta^{SWA} < \sigma^2$, that is, the denominator is one order of $\sigma$ higher than the numerator.

$d\delta^{EWA}/d\sigma$ measures the trend of $\delta^{EWA}$ as $\sigma$ increases:

$$\frac{d\delta^{EWA}}{d\sigma} = \frac{\dfrac{\left(\delta^{EWA}\right)^2}{\sigma^3} - \dfrac{1}{\sigma}}{\left(\dfrac{\delta^{EWA}}{\sigma^2} - \dfrac{1}{I_1 + \bar{I}_2^{-EWA}}\right)} > \frac{\delta^{EWA}}{\sigma} - \frac{\sigma}{\delta^{EWA}} \tag{A4}$$

$d\delta^{EWA}/d\sigma$ is calculated in the same way as the process in Equation (A2), and $\delta^{EWA}$ is larger than $\delta^{SWA}$ based on its definition; hence, $d\delta^{EWA}/d\sigma > d\delta^{SWA}/d\sigma > 0$.

$d^2\delta^{EWA}/d\sigma^2$ measures the trend of $d\delta^{EWA}/d\sigma$ as $\sigma$ increases:

$$\frac{d^2\delta^{EWA}}{d\sigma^2} = \frac{\dfrac{2\delta^{EWA}\left(\delta^{EWA}\right)' \cdot \sigma^3 - 3\left(\delta^{EWA}\right)^2 \sigma^2 + \sigma^4}{(\sigma^3)^2}}{\left(\dfrac{\delta^{EWA}}{\sigma^2} - \dfrac{1}{I_1 + \bar{I}_2^{-EWA}}\right)}$$
$$-\frac{\left(\dfrac{\left(\delta^{EWA}\right)^2}{\sigma^3} - \dfrac{1}{\sigma}\right) \times \dfrac{\left(\delta^{EWA}\right)' \cdot \sigma^2 - 2\sigma \cdot \delta^{EWA}}{(\sigma^2)^2}}{\left(\dfrac{\delta^{EWA}}{\sigma^2} - \dfrac{1}{I_1 + \bar{I}_2^{-EWA}}\right)^2} \tag{A5}$$

Similar to the trend of $d^2\delta^{SWA}/d\sigma^2$, $-1 < d^2\delta^{SWA}/d\sigma^2 < 0$ can be derived.

## Appendix B

The optimality conditions change with different limitations working according to Equation (15).

1.  Without considering $\delta^{min}$.

2.  When $\bar{I}_2^{SWA} < \bar{I}_2 < \bar{I}_2^G$ or $\bar{I}_2^{SWA} < \bar{I}_2 < \bar{I}_2^H$, $f_2(\delta^*)$ is larger than $f_2(\delta^{SWA})$ that exceeds MUHG, i.e., $f_2(\delta^*) > f_2(\delta^{SWA}) = f_1(S_1^A)$ or $f_2(\delta^*) > f_2(\delta^{SWA}) = f_1(S^L)$, allocating as much space as possible or all of the space above FLSV, which depends on the lower bound of carryover storage, to accommodate the relative larger inflows to reduce flood risk. The optimal conditions of R.1 under different lower bounds of carryover storage can be written as Equations (A6) and (A7):

$$\begin{cases} S_1^* = S_1^A \\ \delta^* = Q_{thres} + S^L - \bar{I}_2 - S_1^A \\ f_2(\delta^*) = \lambda \\ f_2(\delta^*) - f_1(S_1^A) = \mu_1^{sl_1} > 0 \\ \mu_1^{su_1} = \mu_1^{su_2} = \mu_1^{sl_2} = \mu_2^\delta = 0 \end{cases} \tag{A6}$$

$$\begin{cases} S_1^* = S^L \\ \delta^* = Q_{thres} - \bar{I}_2 \\ f_2(\delta^*) = \lambda \\ f_2(\delta^*) - f_1(S^L) = \mu_1^{sl_2} > 0 \\ \mu_1^{su_1} = \mu_1^{su_2} = \mu_1^{sl_1} = \mu_2^{\delta} = 0 \end{cases} \tag{A7}$$

which suggests little or no water storage and the shadow price of the lower bound for carryover storage is positive, i.e., $\mu_1^{sl_1} > 0$ or $\mu_1^{sl_2} > 0$.

When $\bar{I}_2^{EWA} \le \bar{I}_2 \le \bar{I}_2^{SWA}$, the price shadow of all the inequality constraints is to be zero. We have $f_1(S_1^*) = f_2(\delta^*)$, and the optimality conditions of R.2 are:

$$\begin{cases} f_2(\delta^*) = f_1(S_1^*) = \lambda \\ \mu_1^{sl_1} = \mu_1^{sl_2} = \mu_1^{su_1} = \mu_1^{su_2} = \mu_2^{\delta} = 0 \end{cases} \tag{A8}$$

When $\bar{I}_2^{EWA} \le \bar{I}_2 \le \bar{I}_2^{SWA}$, we have $f_1(S_1^C) = f_2(\delta^{SWA}) > f_2(\delta^*)$, the shadow price of the upper bound of carryover storage is positive (i.e., $\mu_1^{su_1} > 0$), and the optimality conditions of R.3 are:

$$\begin{cases} S_1^* = S_1^C \\ \delta^* = Q_{thres} + S^L - \bar{I}_2 - S_1^C \\ f_2(\delta^*) = \lambda \\ f_1(S_1^C) - f_2(\delta^*) = \mu_1^{su_1} > 0 \\ \mu_1^{sl_1} = \mu_1^{sl_2} = \mu_1^{su_2} = \mu_2^{\delta} = 0 \end{cases} \tag{A9}$$

3. Case 1

When $\delta^{min}$ is smaller than $\delta^{SWA}$, the optimal conditions are the same as the optimal conditions without considering $\delta^{min}$ since it will not alter the space of hedging and decision making.

4. Case 2

When $\delta^{SWA} \le \delta^{min} \le \delta^{EWA}$ and the forecasted inflow satisfies $\bar{I}_2^M < \bar{I}_2 \le \bar{I}_2^G$ or $\bar{I}_2^M < \bar{I}_2 \le \bar{I}_2^H$, the MUPG exceeds MUFR, i.e., $f_1(S_1^*) > f_1(S_1^M) = f_2(\delta^{min})$, implying that floodwater should be stored after satisfying the minimum flood-safety margin. Under this situation, the shadow price of the minimum safety margin constraint is positive, i.e., $\mu_2^{\delta} > 0$, and the optimality conditions are:

$$\begin{cases} S_1^* = Q_{thres} + S^L - \bar{I}_2 - \delta^{min} \\ \delta^* = \delta^{min} \\ f_1(S_1^*) = \lambda \\ f_1(S_1^*) - f_2(\delta^*) = \mu_2^{\delta} > 0 \\ \mu_1^{su_1} = \mu_1^{su_2} = \mu_1^{sl_2} = \mu_1^{sl_1} = 0 \end{cases} \tag{A10}$$

The optimal solutions of Case 2-R.2 and Case 2-R.3 are consistent with R.2 and R.3, respectively.

5. Case 3

$\delta^{min}$ is larger than $\delta^{EWA}$, illustrating a very small flood risk, and the optimal conditions of Case 3-R.1 and Case 3-R.2 are the same as Equations (A9) and (A10).

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
