# Peer review of "An Analytical Framework for Investigating Trade-Offs between Reservoir Power Generation and Flood Risk"

_water, doi:10.3390/w14233841_

Round 1

Reviewer 1 Report

The manuscript titled-An analytical framework for investigating the tradeoffs between reservoir power generation and flood risk addresses an essential concept of the connections between flood risk and power generation through the derivation of optimal hedging rules.

Overall the manuscript is well-written, with a clear description of the past literature and present challenges (research gaps), workflow, description of the problem, methodology, and results. I have a few minor comments that the authors may want to address to improve the manuscript's readability.

1. A single legend can be used in figure 3.

2. Geographical coordinates of the study area may be added in Figure 9.

3. Line No. 474: What is the reason behind selecting the weight of power generation and preference for flood risk at 0.2 and 0.8, respectively?

4. Figure 12: The maximum flood risk represented for different years. Does it represent the maximum value for a given flood event in a particular year? Please mention this in detail for clarity to the readers of Water.

5. The generic applicability of the proposed framework to other case studies may be highlighted in the conclusions section. How future studies may also incorporate the impacts of climate change may also be mentioned.

6. The definition of flood risk adopted in the study must be mentioned clearly, as terminologies differ in their applicability to a particular flood management problem.

Author Response

We greatly appreciate the constructive comments and detailed reviews. We have carefully revised the manuscript, and we think its readability has improved.

Reviewer 2 Report

Major revision is required

Author Response

We appreciate for Reviewer’s warm work earnestly and hope that the corrections and responses will meet with approval. We have carefully modified the manuscript according to these comments and believe the manuscript has been improved significantly according to the reviewers’ comments.

Round 2

Reviewer 2 Report

Authors have addressed all the raised concern. Now, it can accepted for publication